# Leveraging The Hints: Adaptive Bidding in Repeated First-Price Auctions

**Wei Zhang**[1]    **Yanjun Han**[2]*    **Zhengyuan Zhou**[3,4]*    **Aaron Flores**[5]    **Tsachy Weissman**[6]

[1]MIT EECS    [2]MIT IDSS    [3]Arena Technologies    [4]NYU Stern    [5]Yahoo! Research    [6]Stanford EE

{w_zhang, yjhan}@mit.edu   z@arena-ai.com   aaron.flores@yahooinc.com   tsachy@stanford.edu

## Abstract

With the advent and increasing consolidation of e-commerce, digital advertising has very recently replaced traditional advertising as the main marketing force in the economy. In the past four years, a particularly important development in the digital advertising industry is the shift from second-price auctions to first-price auctions for online display ads. This shift immediately motivated the intellectually challenging question of how to bid in first-price auctions, because unlike in second-price auctions, bidding one's private value truthfully is no longer optimal. Following a series of recent works in this area, we consider a differentiated setup: we do not make any assumption about other bidders' maximum bid (i.e. it can be adversarial over time), and instead assume that we have access to a hint that serves as a prediction of other bidders' maximum bid, where the prediction is learned through some blackbox machine learning model. We consider two types of hints: a single point-prediction, and a hint interval (representing a type of confidence region into which others' maximum bid falls). We establish minimax near-optimal regret bounds for both cases and highlight the quantitatively different behavior between them. We also provide improved regret bounds when the others' maximum bid exhibits the further structure of sparsity. Finally, we complement the theoretical results with demonstrations using real bidding data.

## 1   Introduction

As e-commerce proliferates across the industries, digital advertising has become the predominant marketing force in the economy: in 2019, businesses in the US alone [Wag19] have spent more than 129 billion dollars on digital advertising, surpassing for the first time the combined amount spent via traditional advertising channels by 20 billion dollars. Since then, this number has been growing and continues to outpace traditional advertising spending [Wag19]. Within digital advertising, the key step that generates such revenue for digital advertising is online ads auctions. In the past, second-price auctions, as a result of its truthful nature, have been the standard format for online ads auctions [LR00, Kle04, LRBPR07]. However, the industry has recently witnessed a shift from second-price auctions to first-price auctions in display ads auctions, which currently account for 54% of the digital advertising market* share [DRS19].

This industry-wide shift to first-price auctions has occurred for several reasons, including larger revenue (the exchange charges a percentage of the winning bid) [Ben18a] and no last-look advantage

---

*Correspondence to Yanjun Han (yjhan@mit.edu) and Zhengyuan Zhou (z@arena-ai.com).

*Search auctions dominate the remaining market, which are still conducted using second-price (or generalize second-price) auctions.

36th Conference on Neural Information Processing Systems (NeurIPS 2022).

for exchanges: under second-price auctions, an ad exchange can examine all the submitted bids and *then* raise the floor price to above the second-highest bid and obtain a larger[*] revenue [Ben18b].

Motivated by these considerations, several ad exchanges, including AppNexus (now Xandr), Index Exchange and OpenX, started to roll out first-price auctions in 2017 and completed the transition by 2018 [Slu17, App18]. Google Ad Manager (previously Adx) followed suit and also completed the move to first-price auctions at the end of 2019 [Dav19] and incorporated additional transparency[*] in their new first-price auction platform: bidders would be able to see the minimum-bid-to-win (i.e. full information) after each auction on Google Ad Manager, whereas in many other ad exchanges, a bidder only knows whether he/she wins the bid (i.e. binary feedback) [Dav19]. Situated in this background, an important question arises: how should a bidder adaptively bid in repeated online first-price auctions to maximize the cumulative payoffs?

Prior to the shift, bidding is straightforward: the optimal bidding strategy is simply truthfully bid one's private value (regardless of what the other bidders do). However, this truthful property no longer holds in first-price auctions. As such, bidding in first-price auctions–and the various inference/learning problems arising from it–quickly become complicated. In response, an online decision-making approach has emerged recently, where a bidder at each auction $t$ needs to decide the amount $b_t$ to bid with a given valuation $v_t$, whereas others' maximum bid $m_t$ is either assumed to be iid drawn from a distribution (unrelated to anything else) or fully adversarial. More specifically, [BGM$^+$19] (and its follow-up journal version [BGM$^+$21]) studied the binary feedback setting and show that: 1) if $m_t$ is drawn iid from an underlying distribution (with a generic CDF), then one achieves the minimax optimal regret of $\widetilde{\Theta}(T^{\frac{2}{3}})$; 2) if $m_t$ is adversarial, then one achieves the minimax optimal regret of $\widetilde{\Theta}(T^{\frac{3}{4}})$. Subsequently, [HZW20] considered the winning-bid only feedback (i.e. a bidder can observe the winning bid) and established that if $m_t$ is drawn iid from an underlying distribution (with a generic CDF), one can achieve the minimax optimal regret of $\widetilde{\Theta}(T^{\frac{1}{2}})$. While it remains unknown what the result would be when $m_t$ is adversarial under winning-bid only feedback, [HZF$^+$20] studied the full-information feedback setting and showed that the minimax optimal regret of $\widetilde{\Theta}(T^{\frac{1}{2}})$ can be achieved when $m_t$ is adversarial[*]. [ZKH$^+$21] also studied the full-information feedback setting where it designed and implemented a space-efficient variant of the algorithm proposed in [HZF$^+$20] and through empirical evaluations, showed that the algorithmic variant is quite effective.

In practice, others' highest bid $m_t$ is often neither stochastic nor adversarial, and contextual information is often available to gain some knowledge of $m_t$. We aim to make inroads into this more practical setup by considering a differentiated setup from the existing and growing adaptive-bidding-in-first-price auctions literature: we do not make any assumption about other bidders' maximum bid (i.e. it can be adversarial over time), or model the contexts directly [BFG21]; instead we assume an access to a hint that serves as a prediction of other bidders' maximum bid, where the prediction is learned through some blackbox machine learning model which could be much more powerful than simple linear models. We consider two types of hints: one where a single point-prediction is available, and the other where a hint interval (representing a type of confidence region into which others' maximum bid falls) is available. We establish minimax optimal regret bounds for both cases and highlight the quantitatively different behavior between the two settings. We also provide improved regret bounds when the others' maximum bid exhibits the further structure of sparsity. Finally, we complement the theoretical results with demonstrations using real bidding data.

## 1.1 Additional Application: Personalized Hospitality Pricing

Another important application that shares similar elements to adaptive bidding in first-price auctions and that hence is also amenable to the methodological framework we develop in this paper is personalized hospitality pricing. In personalized hospitality pricing, a travel distribution platform applies a markup to a given hotel room provided by a supplier (either the hotel itself or some travel

---

[*]At the extreme, raising the floor price to barely under the highest bid effectively turns it into a first-price auction without the bidders being aware.

[*]Google was under sustained criticism of leveraging last-look advantage in second-price auctions. This is likely an effort to offset the previous negative image, although there was no such mention in Google's official language.

[*]Note that under both full-information feedback and iid $m_t$, a pure exploitation algorithm already achieves the minimax optimal regret $\Theta(\sqrt{T})$.

aggregator such as Expedia) and presents the final price along with the hotel room whenever a user (either a consumer or a travel agency) searches for a hotel when booking travel. To see the parallel with first-price auctions, the supplier's cost corresponds to the private value, and the final price (which is supplier's cost plus the markup) corresponds to the "bid". Note that, the user can access many other competing travel distribution platforms, each of which may provide a different price for the same hotel room (type). Consequently, there is a bidding element because the user will take the lowest price. Further, in this problem, the platform would want to provision personalized markups, where the markup is decided based on search features (destination city, number of nights, days until first check-in), hotel features (ratings, room types) and other generic features (holiday season, time of the year etc). Note that as of this writing, although almost all existing markup provisioning schemes are fixed business rules that are handcrafted (many "if this feature then that markup" logic statements), an adaptive learning approach – such as the one proposed in this paper, has great applicability in practice. In particular, Arena Technologies[*], a leading enterprise AI solution provider, provides reinforcement learning enabled personalized hospitality pricing. Using publicly available travel pricing data, Arena builds price prediction models that serve as hints, which are in turn used in its real-time adaptive personalization engine. Such a differentiated infrastructure – both in terms of engineering sophistication and learning flexibility – makes it easy for the deployment and testing of our (and any future improved) algorithm, thereby broadening its potential impact.

## 2   Problem Formulation

We study the problem of repeated first-price auction with hint as follows. Consider a time horizon with total length $T$, and there is one round of first-price auction taking place at each time. At the beginning of each round, a bidder observes a particular item and has a private value $v_t \in [0, 1]$ for it. Then, based on her past observations of others' bid and $v_t$ for that round, she bids $b_t \in [0, v_t]$ for that item. The bidder wins the round if and only if $b_t$ is larger than others' highest bid, defined as $m_t$. Under the above settings one could write the instantaneous reward at time $t$ for that bidder:

$$r(b_t; v_t, m_t) = (v_t - b_t) \cdot \mathbb{1}\{b_t \geq m_t\}. \tag{1}$$

Define policy $\pi$ as the overall bidding strategy, which is a sequence of bidding prices $(b_1, b_2, \cdots, b_T)$ under corresponding private value sequence $\{v_t\}$ and others' highest bid sequence $\{m_t\}$, and we use $\mathbb{E}[r(b_t; m_t, v_t)]$ to denote the expected reward under policy $\pi$ while the expectation is taken over the randomness inside randomized policy $\pi$. Then we define the regret under policy $\pi$:

$$\text{Reg}(\pi) = \max_{a \in \mathcal{F}_{\text{L-M}}} \sum_{t=1}^{T} r(a(v_t); m_t, v_t) - \sum_{t=1}^{T} \mathbb{E}[r(b_t; m_t, v_t)], \tag{2}$$

where $a$ denotes a bidding oracle - a map from private values $v_t$ to bidding prices $b_t$, and $a(v_t)$ is the bidding price under oracle $a$.[*] Here $\mathcal{F}_{\text{L-M}}$ is the set of oracles we compete with, which is the set of all 1-Lipschitz and increasing functions from $[0, 1] \to [0, 1]$.

The above settings are similar to that in [HZW20] [HZF+20], and in this work, we include additional information provided to the bidder at each round. The goal is to analyze how the performance of hints may influence the regret bounds in theory. We consider two forms of hint, both of which contain a point estimate $h_t$ of the minimum-bid-to-win $m_t$ at time $t$, and the difference lies in whether a bidder observes a single hint or a hint interval, with the latter defined as a pair $(h_t, \sigma_t)$ satisfying:

$$\mathbb{E}\left[|h_t - m_t|^q\right] \leq \sigma_t^q, \quad t = 1, 2, \cdots, T, \tag{3}$$

where $q$ measures how accurate the error estimation is, namely, as $q$ becomes larger the bidder is more confident that $h_t$ and $m_t$'s difference is smaller than $\sigma_t$. If we consider the extreme case of $q \to \infty$, then $m_t$ is almost surely inside $[h_t - \sigma_t, h_t + \sigma_t]$. Note that (3) always holds regardless of whether the learner observes a hint or a hint interval, and the main difference is that $\sigma_t$ is only revealed in the latter scenario. The bidder's goal is to maximize the cumulative reward for the whole time horizon, and equivalently, to minimize the overall regret. Let $L := \sum_{t=1}^{T} \sigma_t$ be the total error of hints, the learner aims to achieve a small regret adaptive to the unknown quantity $L$: the regret is never larger than the no-hint case even for large $L$, but becomes significantly smaller if $L$ is small.

---

[*]See https://www.arena-ai.com/ for more information.

[*]In the following we may use $r_{t,a}$ to abbreviate $r(a(v_t); m_t, v_t)$.

# 3 Regret Gap Between Single Hint and Hint Interval

In this section we identify two unique features of leveraging the hints in first-price auctions. First, the best way to use hints is different from the ones in the literature of online learning: instead of adding an optimistic term based on the hints in the multiplicative weights algorithm, in first-price auctions we should manually add the hints as a new expert. Second, first-price auctions exhibit a provable gap between the regrets under single hints and hint intervals, a phenomenon which does not occur in the hint literature.

## 3.1 Online Learning with Hints

There is a rich line of literature related to online learning with hints where one aims to achieve data-dependent regret bounds in terms of the variation in the environment [AAGO06, HK11, CYL$^+$12, RS13, SL14, WL18, BLLW19], by taking part of the revealed past losses implicitly as the hint. The algorithm to leverage these hints typically falls into the category of optimistic online mirror descent, where the hint is used to form an optimistic estimate of the current loss. A piece of work that explicitly formulated the hint similar to ours is [WLA20], which used an optimistic EXP4 algorithm to achieve a regret bound of $O(\min\{\sqrt{T}, \sqrt{L}T^{1/4}\})$ when the total error $L$ of the loss predictors is known.

The nature of first-price auction problems, however, is different from the above. The crucial feature is the discontinuity in the reward function (1) when $b_t$ is close to the minimum-bid-to-win $m_t$, which means that even an accurate prediction of $m_t$ does not imply an accurate reward prediction under every bid $b_t$. This distinction not only leads to different optimal regrets, but also results in different algorithms for regret minimization, as well as a curious gap between the optimal regrets under single hints and hint intervals. The concept of the hint interval does not offer additional help over single hints in many other online learning problems, and to the best of our knowledge, is new in the literature.

## 3.2 Regret Gap Between a Single Hint and a Hint Interval

In this section, we show that the above distinctions are already present in a toy example where all private values $v_t$ are the same for $t = 1, 2, \cdots, T$ (say $v_t \equiv 1$). Our first result states that even under this very simple case, there is a strict separation between the regret bounds of knowing single hints and that of knowing hint intervals.

**Theorem 1.** *For $L \in [1, T]$, $q \in [1, \infty)$, if $v_t \equiv 1$ and the bidder observes a hint interval at each time $t$, the policy $\pi_1$ in Algorithm 1 satisfies*

$$\text{Reg}(\pi_1) \leq C\sqrt{\log T \cdot T^{\frac{1}{q+1}} \cdot L^{\frac{q}{q+1}}},$$

*for a numerical constant $C$ independent of $\{m_t, h_t, \sigma_t\}$. Moreover, the following minimax lower bound holds:*

$$\inf_{\pi} \sup_{\{m_t, h_t, \sigma_t\}} \text{Reg}(\pi) \geq c\sqrt{T^{\frac{1}{q+1}} \cdot L^{\frac{q}{q+1}}}.$$

*Here $c > 0$ is a numerical constant independent of $(T, L)$, the supremum is taken over all $\{m_t, h_t, \sigma_t\}$ sequences that satisfy (3), and the infimum is taken over all possible policies $\pi$.*

**Theorem 2.** *If $v_t \equiv 1$ and the bidder observes a single hint at each time $t$, then for every $q \in [1, \infty)$, the policy $\pi_2$ in Algorithm 1 with $T$ extra experts bidding $h_t + \{0, 1/T, \cdots, 1 - 1/T\}$ achieves*

$$\text{Reg}(\pi_2) \leq C (\log T)^{\frac{1}{2}} (T \cdot L)^{\frac{1}{4}},$$

*for a numerical constant $C$ independent of $\{m_t, h_t\}$. Moreover, the following minimax lower bound holds:*

$$\inf_{\pi} \sup_{\{m_t, h_t\}} \text{Reg}(\pi) \geq c (T \cdot L)^{\frac{1}{4}}.$$

*Here $c > 0$ is a numerical constant independent of $(T, L)$, the supremum is taken over all $\{m_t, h_t\}$ sequences that satisfy (3), and the infimum is taken over all possible policies $\pi$.*

The above result shows that there is a strict separation in regret bounds compared to the previous subsection when additional information is given. Observe that in Theorem 1, when $q$ becomes larger, the minimax regret becomes smaller because the error estimation is more accurate. In Theorem 2, however, the lower bound stays the same order as $q$ changes, and is strictly larger than the case of knowing the hint interval as long as $q > 1$. When $q \to \infty$, the upper and lower bound in Theorem 1 gives an optimal $\widetilde{O}(\sqrt{L})$ magnitude for the minimax regret.

The intuition behind this separation can be explained as follows: hint intervals can be considered as single hints plus an additional information of its accuracy. For example, if the hint interval has length $\epsilon \to 0$, one strategy would be to bid exactly as the hint suggests; however, if the bidder do not observe an interval, it is hard for her to wisely arrange the weight she put in the hints given and thus leading to a smaller reward compared to previous case. This distinction turns out to be crucial because of the discontinuity of the reward in (1).

Also note that the upper and lower bounds in Theorems 1 and 2 are tight within logarithmic factors, and exhibit the desired adaptive regret in the hint performance $L$: when $L$ is as large as $T$, an $\widetilde{O}(\sqrt{T})$ regret is attainable which is optimal without the hints; as the quality of the hints becomes better, the regret dependence on $T$ is greatly improved.

### 3.3 Algorithm for Section 3.2

It is a classical result in online learning that the multiplicative weights algorithm in [LW94] leads to a classical regret upper bound of $O\left(\sqrt{T \cdot \log K}\right)$, for $K$ experts and time horizon $T$ under the setting of prediction with expert advice. With hints in first-price auctions, instead of using optimistic online mirror descent in the literature, we modify the multiplicative weights in another way: we run the same multiplicative weights algorithm with an additional expert in the existing set of experts, whose bid depends on the hint $h_t$.

Specifically, the algorithm to achieve the upper bound in Theorem 1 is Algorithm 1. Construct $T$ base experts, and let the $i$-th ($i = 1, 2, \ldots, T$) base expert bid $\frac{i}{T}$ at each time $t$ (since $v_t \equiv 1$ the Lipschitz oracle will bid a constant value). It is easy to see that the discretization error incurs an additional regret at most $T \cdot (1/T) = \Theta(1)$. Now with the hint interval, we include an extra expert $a^*$ who bids $h_t + \sigma_t^{q/(q+1)}$ at each time $t$. Consequently, we have a set of $K = T + 1$ experts in total, containing $T$ base experts and one hint expert. The multiplicative weights algorithm is then applied as follows:

$$ p_{t,a} = \frac{\exp\left(\eta_t \cdot \sum_{s<t} r_{s,a}\right)}{\sum_{a' \in [K]} \exp\left(\eta_t \cdot \sum_{s<t} r_{s,a'}\right)}, \quad a \in [K], \quad t = 1, \ldots, T, $$

The main ideas of our algorithm are:

- We discretize the bids $b_t$ (and possibly the private values $v_t$ when we do not assume that $v_t \equiv 1$ later) when constructing oracles without large loss of cumulative rewards compared to continuous ones, since an enumeration of all 1-Lipschitz functions is unrealistic.

- Imagine if the bidder knew the possible range of $m_t$ at the beginning of time $t$: $[\underline{m}_t, \overline{m}_t]$, then she could bid the upper point $\overline{m}_t$, and this is a $(\overline{m}_t - \underline{m}_t)$-good expert in the sense of [HZF+20, Lemma 1], which is an appropriate tool to handle the discontinuity in the rewards.

We show in Appendix B.1.1 that Algorithm 1 achieves the regret upper bound shown in Theorem 1. As for Theorem 2 we include $N$ hint experts, each of which bids a constant gap $\Delta_i$ above $h_t$ at all time, i.e. the first one bids $b_t = h_t$ for all $t$, and the second one bids $b_t = h_t + \frac{1}{T}$ for all $t$, ect. We now have a "dense" set of hint experts covering all strategies that bid $b_t = h_t + c$, $c \in [0, 1]$, $t = 1, 2, \cdots, T$, with total loss at most $\Theta(1)$. (See details of the proof in Appendix B.1.2.)

Notice that in both cases we required knowledge of $L$ to decide learning rate $\eta_t$ at each time $t$, but this problem can be addressed by existing techniques [ACBG02, YEYS04]. Indeed, the upper bound in Theorem 2 still holds even if $L$ is not known in advance.

---

**Algorithm 1:** Multiplicative Weights with Hint Intervals

---

**Input:** Time horizon $T$; Hints accuracy $L$; $q$.
**Output:** A bidding policy $\pi$.
**Initialization:** Construct $K = T + 1$ experts, with the $i$-th ($i = 1, 2, \ldots, T$) base expert bidding
$\frac{i}{T}$ at each time $t$ and an extra expert $a^*$ bidding $h_t + \sigma_t^{q/(q+1)}$ at each time $t$;

**for** $a \in [K]$ **do**
  | Initialize $r_{s,a} \leftarrow 0$;
**end**
**for** $t \in \{1, 2, \cdots, T\}$ **do**
  | The bidder receives a private value $v_t \equiv 1$;
  | The bidder observes hint $h_t \in [0, 1]$, along with its accuracy $\sigma_t$;
  | Set $b_{t,a^*} \leftarrow h_t + \sigma_t^{\frac{q}{q+1}}$;
  | Set learning rate $\eta_t \leftarrow \min \left\{ \frac{1}{4}, \sqrt{\frac{\log K}{\sum_{s \leq t} \sigma_s^{\frac{q}{q+1}}}} \right\}$;
  | Let $b_t \leftarrow b_{t,a}$ with probability

  $$p_{t,a} = \frac{\exp \left( \eta_t \cdot \sum_{s < t} r_{s,a} \right)}{\sum_{a' \in [K]} \exp \left( \eta_t \cdot \sum_{s < t} r_{s,a'} \right)}.$$

  | The bidder receives others' highest bid $m_t$;
  | **for** $a \in [K]$ **do**
  |   $$r_{t,a} \leftarrow r_{t,a} + r(b_{t,a}; v_t, m_t).$$
  | **end**
  | Update $R_t \leftarrow R_{t-1} + r(b_t; v_t, m_t)$;
**end**

---

### 3.4 Varying Private Prices

When private values could vary, oracles cannot bid the same price anymore. The following theorem holds, again showing a strict separation and furthermore indicates that for varying $v_t$, hints does not help much for continuous bidding value.

**Theorem 3.** *Let $L \in [1, T]$, $q \in [1, \infty)$, and $v_t \in [0, 1]$ for $t = 1, 2, \ldots, T$. If the bidder observes hint intervals, the following characterization of the minimax regret holds:*

$$\inf_{\pi} \sup_{\{v_t, m_t, h_t, \sigma_t\}} \mathrm{Reg}(\pi) = \widetilde{\Theta} \left( \min \left\{ T^{\frac{1}{q+1}} L^{\frac{q}{q+1}}, \sqrt{T} \right\} \right),$$

*where the supremum is taken over rewards under all possible $\{v_t, m_t, h_t, \sigma_t\}$ sequences and hints that satisfy (3), and the infimum is taken over all possible policies $\pi$.*

**Theorem 4.** *Let $L \in [1, T]$, and $v_t \in [0, 1]$ for $t = 1, 2, \ldots, T$. If the bidder observes single hints, then $\forall q \in [1, \infty)$, the following characterization of the minimax regret holds:*

$$\inf_{\pi} \sup_{\{v_t, m_t, h_t, \sigma_t\}} \mathrm{Reg}(\pi) = \widetilde{\Theta} \left( \sqrt{T} \right),$$

*where the supremum is taken over rewards under all possible $\{v_t, m_t, h_t, \sigma_t\}$ sequences and hints that satisfy (3), and the infimum is taken over all possible policies $\pi$.*

Although both results in Theorems 3 and 4 are tight within logarithmic factors, they are pessimistic results. When the hint intervals are observed, Theorem 3 shows that the help from the hint exhibits a thresholding phenomenon: either bidding without hints or only bidding the hints is optimal. For example, when $q = \infty$, the minimax regret is simply a tedious quantity $\widetilde{\Theta}(\min\{L, \sqrt{T}\})$. When there are only single hints, Theorem 4 even shows that the hint is of no help unless its quality is very high, i.e. $L \leq 1$. This pessimistic situation is alleviated in the next section, by imposing an additional assumption that others' bids $m_t$ are only supported on a few locations.

# 4 Exploiting the Sparsity of Others' Bids

The previous section gives a pessimistic result that hints help only to the same extent of bidding the hint itself. To mitigate this drawback, we identify a useful structure in practical online first-price auctions: the maximum competing bids $m_t$ are supported on only a few locations. See Figure 1 for a typical example in real data. The reason why sparsity arises in practice is partially due to the scenario where the maximum competing bid is the reserve price set by the seller.

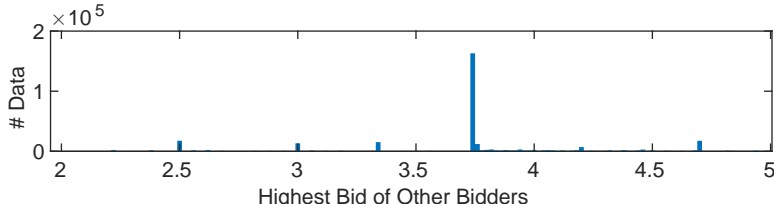

Figure 1: Histogram of others' highest bid in real data.

Assuming that $m_t$ is only supported on $K$ locations, the central question in this section is as follows:

*How does the sparsity improve the minimax regret? Can we devise a learning algorithm that is adaptive to the parameter $K$ (and other parameters such as $L$)?*

## 4.1 Minimax Regret with Sparsity

The central result of this section is the following characterization of the minimax regret with sparsity:

**Theorem 5.** *For $q \in [1, \infty)$ and varying private prices, suppose that the minimum-bid-to-win $m_t$ only takes $K$ support values and the hint intervals are available. The regret is upper bounded by:*

$$\inf_{\pi} \sup_{\{v_t, m_t, h_t, \sigma_t\}} \text{Reg}(\pi) = O\left(\min\left\{\sqrt{\log T \cdot T^{\frac{1}{q+1}} \cdot L^{\frac{q}{q+1}} \cdot K}, T^{\frac{1}{q+1}} \cdot L^{\frac{q}{q+1}}, \sqrt{T}\right\}\right),$$

*where the supremum is taken over all $\{v_t, m_t, h_t, \sigma_t\}$ sequences and hints that satisfies (3), and the infimum is taken over all possible policies $\pi$. In addition, the following minimax lower bound for the regret holds:*

$$\inf_{\pi} \sup_{\{v_t, m_t, h_t, \sigma_t\}} \text{Reg}(\pi) = \Omega\left(\min\left\{\sqrt{T^{\frac{1}{q+1}} \cdot L^{\frac{q}{q+1}} \cdot K}, T^{\frac{1}{q+1}} \cdot L^{\frac{q}{q+1}}, \sqrt{T}\right\}\right).$$

Theorem 5 shows that the minimax regret exhibits an elbow with respect to the sparsity: the regret grows sublinearly with $K$ for small $K$, but reduces to the fixed value $\Theta\left(\min\left\{T^{\frac{1}{q+1}} L^{\frac{q}{q+1}}, \sqrt{T}\right\}\right)$ of Theorem 3 when $K$ is large. Moreover, the better the hint quality is, the more helpful the sparsity will be. For example, if $L = \Theta(1)$, sparsity helps reduce the regret whenever $K < T^{1/(q+1)}$; as the other extreme, if $L = \Theta(T)$, the classical $\Theta(\sqrt{T})$ regret is always unavoidable, no matter how small $K$ is.

## 4.2 Meta Algorithm Adaptive to Unknown $L$ and $K$

In the proof of Theorem 5 (see appendix C.1) we prove the three upper bounds can indeed be achieved separately. If a bidder observes the overall quality of hints $L$ at the beginning, she can easily choose the best among those three bounds and achieve an optimal. It is not straightforward though, to achieve them simultaneously without this knowledge. For instance, to achieve each single upper bound, there is a data-driven algorithm agnostic to $L$ (such as using the idea in [ACBG02]); however, the choice of which algorithm to use still depends on the knowledge of $L$.

Algorithm 2 (pseudocode in Appendix A) with meta-experts addresses this problem and ensures to achieve the optimal one among the three upper bounds for any $L = \Omega(1)$. The intuition is to give hints a higher priority if the error is small. Meanwhile, with this meta structure, it is possible to set different fine tuning strategy for the two layers. By including three "Meta Experts" in the upper layer,

whose strategies are indeed output of algorithms rather than pre-designed oracles, we can achieve the minimum of the three regret bounds listed in Theorem 5.

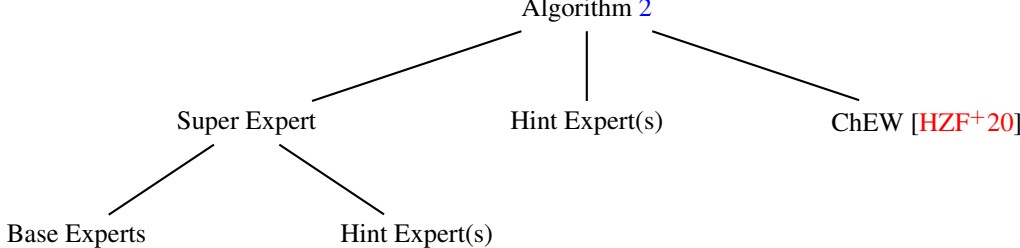

**Theorem 6.** *Algorithm 2 achieves the regret upper bound of Theorem 5, while it is agnostic to the knowledge of L.*

Theorem 6 is also true when the support size $K$ and support locations are unknown together with $L$.[*] Using similar ideas of the doubling trick, we can deal with unknown $K$. At the beginning assume a constant value $K_0$ (e.g. $K_0 = 8$), and run the above algorithm until the number of observed supports up to time $t$ exceeds the current value. If that happens, we double the value of $K$: $K_{i+1} = 2 \cdot K_i$ ($i \in \mathbb{N}$), and restart the learning algorithm. Since assuming a larger $K$ than reality only makes the oracle stronger, substituting the regret upper bound (4) below to this strategy leads to regret upper bound (let $M := \lceil \log \left( \frac{K}{8} \right) \rceil + 1$):

$$O \left( \sqrt{\log T} \cdot \sum_{i=0}^{M-1} \sqrt{K_i \cdot \sum_{t=T_i}^{T_{i+1}} \sigma_t^{\frac{q}{q+1}}} \right) \leq O \left( \sqrt{\log T} \cdot \sqrt{(K_0 + \ldots + K_M) \cdot \left( \sum_{t=1}^{T} \sigma_t^{\frac{q}{q+1}} \right)} \right)$$

$$\leq O \left( \sqrt{\log T \cdot 2K \cdot T^{\frac{1}{q+1}} \cdot L^{\frac{q}{q+1}}} \right),$$

where $K_i := K_0 \cdot 2^i$ and $T_i$ are the dividing points when current number of supports exceeds current $K$ ($T_{M+1} = T$).

For unknown support values construct an expert set as follows. We divide $y$-axis (bidding price) into $T$ small intervals, with the $i$-th interval being $\left[ \frac{i}{T}, \frac{i+1}{T} \right]$ ($i = 0, 1, \cdots, T - 1$). Our goal is to allocate the $K$ supports into these $T$ intervals, with each interval containing at most one support (two supports in the same interval could be merged with an additional $O(1)$ regret). The total number of allocations is $\binom{T}{K}$, and for each allocation, the oracle can only choose from $T^K$ bidding functions (cf. the proof of Theorem 5), thus the size of the complete expert set is:

$$|\text{expert set}| \leq \binom{T}{K} \cdot T^K \leq T^{2K}.$$

Then the regret upper bound can be written as (using similar analysis to Appendix B.1.1)

$$O \left( \sqrt{\log(T^{2K}) \cdot \sum_{t=1}^{T} \sigma_t^{\frac{q}{q+1}}} \right) = O \left( \sqrt{K \cdot \log T \cdot \sum_{t=1}^{T} \sigma_t^{\frac{q}{q+1}}} \right). \tag{4}$$

#### 4.2.1 Single Hints Case

In Section 3 we mainly showed that there is a strict difference in the regret bound for single hints and hint intervals case. Here we also provide results for single hints case for completeness.

**Theorem 7.** *For $q \in [1, \infty)$ and varying private prices, if the minimum bid to win takes $K$ support values and the bidder only observes a single hint at each time then the regret bounds hold:*

$$\inf_{\pi} \sup_{\{v_t, m_t, h_t, \sigma_t\}} \text{Reg}(\pi) = \widetilde{\Theta} \left( \min \left\{ \sqrt{T}, \sqrt{\sqrt{LT} \cdot K} \right\} \right),$$

*where the supremum is taken over all $K$-sparse $m_t$ sequence and hints that satisfies (3), and the infimum is taken over all possible policies $\pi$.*

---

[*] Since our algorithm is adaptive to $T$ as discussed in the previous section, it is equivalent to say no parameter about the whole game is known.

# 5 Real-data Experiments

This section presents several real-data experiments in repeated first-price auctions based on practical bidding data, where the hint is the context-based prediction provided by blackbox machine learning models. For business confidentiality we do not disclose further information about the datasets.

Our experiments are run on auction datasets from the first-price auctions on real-world sites, with around 0.38 million data points, where each data point is a quadruple of scalars $(v_t, m_t, h_t, \sigma_t)$. Here $v_t$ is the private value and $m_t$ is the minimum bid to win for each auction. The private value $v_t$ is computed by Verizon Media based on an independent learning scheme not relying on the auction, and is therefore taken as given. The quantity $m_t$ is the minimum bid to win and is returned by the platform after each auction, which is by definition the other bidders' highest bid (possibly including the seller's reserve price and measured up to 1 cent). These datasets have already been pruned to only contain data points with $v_t > m_t$, for otherwise the bidder never wins regardless of her bids. Hints $h_t$ and its accuracy $\sigma_t$ are provided by fitting a lognormal model using other contextual information.

We did two parts of experiments. In the first part, we applied the main insight in Section 3 that for fixed private value, given extra information of hints helps reduce the regret for optimal policy, and hint intervals outperforms single hints, as long as $q > 1$. We allocate all data points to separate bins according to the private value $v_t$ and each bin is a subproblem described in Section 3. Figure 2(a) compares the results of binned exponential weighting (without hint) and whether hint intervals or single hints are given. We can observe although the performance of hints itself is obviously bad, by incorporating hint into the learning method, we could improve the cumulative reward by 2.96% if knowing point estimation only, and by 3.55% if an interval is provided in each round.

We also provide a polynomial time algorithm for conducting the meta algorithm we proposed before. Even with $T^K$ experts, we may use dynamic programming to achieve a space complexity $O(DK)$ and time complexity $O(DKT)$, where $D$ is parameter for the discretization on private values. We implemented Algorithm 3 on dataset with support size $K = 5$.

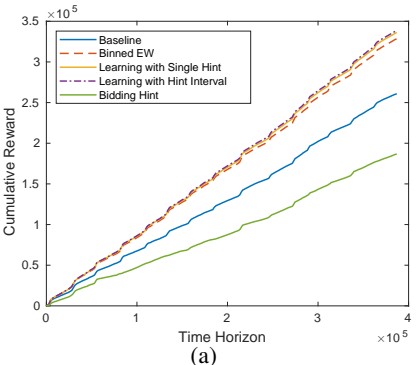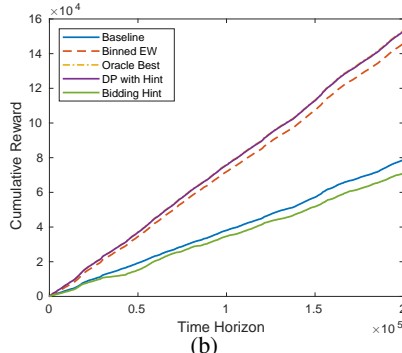

Figure 2: Cumulative rewards as a function of time, based on Algorithms 1 and 3. Panel (a) uses a binning method to show the improvement by incorporating hints. The yellow solid line corresponds to learning with single hints, while the dashdot line corresponds to hint intervals. As comparisons, we also include a baseline algorithm (blue) provided by Verizon Media, the simple exponential weights algorithm without hints (red), and just bidding the hint (green). Panel (b) illustrates the performance of our Algorithm 3 (purple solid line), which almost coincides with the performance of the best increasing and 1-Lipschitz function for the oracle (yellow dashed line). The other algorithms are the same as Panel (a).

# 6 Conclusion

In this work we study the overall regret for a particular bidder without further assumption for other bidders in repeated first-price auction. We target at the case when an additional information is given to only this bidder at each round. We show that even in a simple setting, there is a strict gap between two different forms of hints, where in either one the upper bound and lower bound matches with regard to log factor. We further consider the case when others' highest bid lies in finite support,

and provide modified algorithm as well as matching regret upper and lower bounds. While not knowing the critical parameters for the whole game in the beginning, our adaptive algorithm always achieves the best among three different upper bounds. Finally, we appreciate data from company that corresponds with our framework, and carry out two parts of experiments on it. The first one is mainly to show the performance improvement with the information given, and in the second part, we provide a way to implement our algorithm in polynomial time and show corresponding results.

## 7 Acknowledgements

This work was supported in part by NSF awards CCF-2106467 and CCF-2106508. Zhengyuan Zhou acknowledges the generous support from New York University's Center for Global Economy and Business faculty research grant. We would like to thank the NeurIPS 2022 reviewers for their constructive feedback and Eric Ordentlich for helpful discussions in the early stage of this work.

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
