# A Pseudocode of Algorithm 2

---

**Algorithm 2:** Meta-Expert Learning Algorithm

---

**Input:** Time horizon $T$; support size $K$; accuracy of error information $(\sigma_1, \cdots, \sigma_T)$; norm parameter $q \in [1, \infty]$.

**Output:** A bidding policy $\pi$.

**Initialization:** Construct $T^K$ base experts $\{f_i\}$ $(i = 1, 2, \cdots, T^K)$ that cover the oracle with cumulative reward difference at most $O(1)$ (as in the proof of Theorem 5);

**for** $i = 1, 2, \cdots T^K$ **do**
  | Initialize $R_{0, f_i} \leftarrow 0$;
**end**

Initialize $R_{0,h} \leftarrow 0$, $R_{0,g} \leftarrow 0$, $R_{0,f} \leftarrow 0$;

Initialize $L_0 \leftarrow 0$ ;

**for** $t \in \{1, 2, \cdots, T\}$ **do**

  The bidder receives private value $v_t \in [0, 1]$;

  Set learning rate $\eta_{t,1} \leftarrow \min\left\{\frac{1}{4}, \sqrt{\frac{K \log T}{L_t}}\right\}$;

  The bidder observes hint $h_t \in [0, 1]$, along with its accuracy $\sigma_t$;

  $L_t \leftarrow L_{t-1} + \sigma_t^{\frac{q}{q+1}}$;

  Set $b_{t,h} \leftarrow h_t + \sigma_t^{\frac{q}{q+1}}$;

  Sample $b_{t,g}$ according to ChEW policy;

  **for** $i = 1, 2, \cdots, T^K$ **do**
    | Let $b_{t,f} \leftarrow f_i(v_t)$ with probability

$$p_{t,i} := \frac{\exp\left(\eta_{t,1} R_{t-1, f_i}\right)}{\exp\left(\eta_{t,1} R_{t-1,h}\right) + \sum_{i'=1}^{T^K} \exp\left(\eta_{t,1} R_{t-1, f_{i'}}\right)}.$$

  **end**

  Let $b_{t,f} \leftarrow b_{t,h}$ with probability $p_{t, T^K+1} := \frac{\exp(\eta_{t,1} R_{t-1,h})}{\exp(\eta_{t,1} R_{t-1,h}) + \sum_{i'=1}^{T^K} \exp\left(\eta_{t,1} R_{t-1, f_{i'}}\right)}$;

  Sample $b_{t,f} \sim p_t$ ;

  Set learning rate $\eta_{t,2} \leftarrow \min\left\{\frac{1}{4}, \sqrt{\frac{\log 3}{L_t}}\right\}$;

  **for** $i \in \{f, g, h\}$ **do**

$$P_{t,i} = \frac{\exp\left(\eta_{t,2} R_{t-1,i}\right)}{\sum_{i' \in \{f,g,h\}} \exp\left(\eta_{t,2} R_{t-1,i'}\right)};$$

  **end**

  The bidder samples policy $i \sim P_t$ and bids $b_{t,i}$;

  The bidder receives others' highest bid $m_t$;

  **for** $i = 1, 2, \cdots, T^K$ **do**

$$R_{t, f_i} \leftarrow R_{t-1, f_i} + r(f_i(v_t); v_t, m_t).$$

  **end**

  **for** $i \in \{f, g, h\}$ **do**
    | Update $R_{t,i} \leftarrow R_{t-1,i} + r(b_{t,i}; v_t, m_t)$;
  **end**

**end**

---

The algorithm has a tree structure with the nodes in the upper layer representing algorithms instead of specific oracles. In Algorithm 2, the upper nodes are respectively: the algorithm that achieves the regret upper bound in Theorem 5 described in Appendix C.1, "ChEW" algorithm to achieve $\widetilde{O}(\sqrt{T})$ regret bound proposed in [HZF+20], and a single expert which bids $h_t + \sigma_t^{q/(q+1)}$ each time. The probability distribution $P_{t,i}$ runs the multiplicative weights update on the above strategies (see details in Appendix C.2).

# B  Proof of Main Result in Section 3

## B.1  Proof of Regret Upper Bounds in Theorem 1 and Theorem 2

### B.1.1  Proof of Upper Bound in Theorem 1.

We prove a slightly stronger result than Theorem 1:

**Lemma 1.** *If $v_t \equiv 1$ and the bidder observes $\sigma_t$ at each time $t$, then the following regret upper bound holds for Algorithm 1:*

$$\sup_{\{m_t, h_t, \sigma_t\}} \text{Reg}(\pi_1) = O\left(\log T + \sqrt{\log T \cdot \sum_{t=1}^{T} \sigma_t^{\frac{q}{q+1}}}\right),$$

*with $\text{Reg}(\pi)$ defined in (2), and the supremum is taken over all $m_t$ sequences and hints that satisfy (3), and the infimum is taken over all possible policies $\pi$.*

*Proof.* The following is similar to proof of Theorem 3 in [HZF+20]. As in the standard analysis of multiplicative weights [CBL06], define:

$$\phi_t = \frac{1}{K} \sum_{a=1}^{K} \exp\left(\eta_t \cdot \sum_{s<t} r_{s,a}\right), \quad t = 1, \ldots, T+1.$$

Recall that $K = T + 1$ and $a^*$ is the extra expert. We translate every $r_{t,a}$ by $-r_{t,a^*}$ to ensure that $r_{t,a} \in [-1, 1]$ and $r_{t,a^*} = 0$. Then for $t \in [T]$, Jensen's inequality with $\eta_t/\eta_{t+1} \geq 1$ gives

$$(\phi_{t+1})^{\frac{\eta_t}{\eta_{t+1}}} = \left[\frac{1}{K} \sum_{a=1}^{K} \exp\left(\eta_{t+1} \cdot \sum_{s<t+1} r_{s,a}\right)\right]^{\frac{\eta_t}{\eta_{t+1}}}$$

$$\leq \frac{1}{K} \sum_{a=1}^{K} \left[\exp\left(\eta_{t+1} \cdot \sum_{s<t+1} r_{s,a}\right)^{\frac{\eta_t}{\eta_{t+1}}}\right]$$

$$= \phi_t \sum_{a=1}^{K} p_{t,a} \cdot \exp\left(\eta_t \cdot r_{t,a}\right) =: \phi_t \mathbb{E}[\exp\left(\eta_t X_t\right)].$$

Here $X_t$ is a random variable that takes value $r_{t,a}$ with probability $p_{t,a}$. Now using Bernstein's inequality

$$\mathbb{E}[\exp(\lambda X)] \leq \exp\left(\lambda \mathbb{E}[X] + (e^\lambda - \lambda - 1)\text{Var}(X)\right),$$

with $|X - \mathbb{E}[X]| \leq 1$ almost surely, we have

$$\frac{\log \phi_{t+1}}{\eta_{t+1}} - \frac{\log \phi_t}{\eta_t} \leq \mathbb{E}[X_t] + \frac{e^{\eta_t} - \eta_t - 1}{\eta_t} \text{Var}(X_t) \leq \mathbb{E}[X_t] + \eta_t \text{Var}(X_t),$$

where the last inequality is due to $e^x - x - 1 \leq x^2$ for $x \in [0, 1]$. Define $r_t^* := \max_{a \in [K]} r_{t,a}$, we have

$$\text{Var}(X_t) \leq \mathbb{E}[(r_t^* - X_t)^2] \leq 1 \cdot \mathbb{E}[r_t^* - X_t] = r_t^* - \mathbb{E}[X_t].$$

By telescoping and defining $\eta_{T+1} := \eta_T$,

$$\frac{\log \phi_{T+1}}{\eta_T} = \sum_{t=1}^{T} \left[\frac{\log \phi_{t+1}}{\eta_{t+1}} - \frac{\log \phi_t}{\eta_t}\right] \leq \sum_{t=1}^{T} \mathbb{E}[X_t] + \sum_{t=1}^{T} \eta_t \left(r_t^* - \mathbb{E}[X_t]\right). \tag{5}$$

For the left-hand side of (5), we also have

$$\log \phi_{T+1} \geq \eta_T \cdot \max_{a \in [K]} \sum_{s=1}^{T} r_{t,a} - \log K. \tag{6}$$

Combining (5) and (6),

$$\max_{a \in [K]} \sum_{t=1}^{T} r_{t,a} \leq \frac{\log K}{\eta_T} + \sum_{t=1}^{T} (1 - \eta_t) \cdot \mathbb{E}[X_t] + \sum_{t=1}^{T} \eta_t \cdot r_t^*. \tag{7}$$

Rearranging (7) leads to the following upper bound on the cumulative regret:

$$\max_{a \in [K]} \sum_{t=1}^{T} r_{t,a} - \sum_{t=1}^{T} \mathbb{E}[X_t] \leq \frac{\log K}{\eta_T} + \sum_{t=1}^{T} \eta_t r_t^* - \sum_{t=1}^{T} \eta_t \cdot \mathbb{E}[X_t]. \tag{8}$$

Let $V_T := (\log K)/\eta_T + \sum_{t=1}^{T} \eta_t r_t^*$, it remains to upper bound the last term of (8). To do so, note that (7) holds for any intermediate value of $t \in [T]$ as well. Since $\max_{a \in [K]} \sum_{t=1}^{T} r_{t,a} \geq \sum_{t=1}^{T} r_{t,a^*} = 0$, for every $t \in [T]$ we have

$$S_t := \sum_{s=1}^{t} (1 - \eta_s) \cdot \mathbb{E}[X_s] \geq -\frac{\log K}{\eta_{t+1}} - \sum_{s=1}^{t} \eta_s \cdot r_s^* = -V_t \geq -V_T,$$

where the last inequality is due to $\eta_{t+1} \geq \eta_T$ and $r_t^* \geq r_{t,a^*} = 0$ for every $t \in [T]$. Consequently,

$$-\sum_{t=1}^{T} \eta_t \cdot \mathbb{E}[X_t] = -\sum_{t=1}^{T} (S_t - S_{t-1}) \cdot \frac{\eta_t}{1 - \eta_t}$$

$$= -\sum_{t=1}^{T-1} S_t \cdot \left( \frac{\eta_t}{1 - \eta_t} - \frac{\eta_{t+1}}{1 - \eta_{t+1}} \right) - S_T \cdot \frac{\eta_T}{1 - \eta_T}$$

$$\leq V_T \sum_{t=1}^{T-1} \left( \frac{\eta_t}{1 - \eta_t} - \frac{\eta_{t+1}}{1 - \eta_{t+1}} \right) + V_T \cdot \frac{\eta_T}{1 - \eta_T}$$

$$= \frac{V_T \eta_1}{1 - \eta_1} \leq V_T,$$

where we have used that $1/4 \geq \eta_1 \geq \eta_2 \geq \ldots \geq \eta_T > 0$. Plugging this inequality back into (7) gives

$$\max_{a \in [K]} \sum_{t=1}^{T} r_{t,a} - \sum_{t=1}^{T} \mathbb{E}[X_t] \leq 2V_T. \tag{9}$$

Finally it remains to upper bound $\mathbb{E}[V_T]$, where the expectation is taken with respect to the randomness in the hint sequence $\{h_t\}_{t=1}^{T}$. Since the definition of the expert $a^*$ gives that

$$r_t^* \leq (1 - m_t) - (1 - h_t - \sigma_t^{q/(q+1)}) \mathbb{1}(h_t + \sigma_t^{q/(q+1)} \geq m_t)$$

$$\leq \begin{cases} h_t + \sigma_t^{q/(q+1)} - m_t & \text{if } h_t + \sigma_t^{q/(q+1)} \geq m_t \\ 1 & \text{if } h_t + \sigma_t^{q/(q+1)} < m_t \end{cases},$$

we conclude that

$$\mathbb{E}[r_t^*] \leq \mathbb{P}(h_t + \sigma_t^{q/(q+1)} < m_t) + \mathbb{E}[|h_t + \sigma_t^{q/(q+1)} - m_t|]$$

$$\leq \frac{\mathbb{E}[|h_t - m_t|^q]}{(\sigma_t^{q/(q+1)})^q} + (\mathbb{E}[|h_t - m_t|^q])^{1/q} + \sigma_t^{q/(q+1)}$$

$$\leq 2\sigma_t^{q/(q+1)} + \sigma_t \leq 3\sigma_t^{q/(q+1)}.$$

Therefore,

$$\mathbb{E}[V_T] \leq \frac{\log K}{\eta_T} + \sum_{t=1}^{T} \eta_t \mathbb{E}[r_t^*]$$

$$\leq 4 \log K + \sqrt{\sum_{t=1}^{T} \sigma_t^{q/(q+1)} \log K} + 3 \sum_{t=1}^{T} \sqrt{\frac{\log K}{\sum_{s \leq t} \sigma_s^{q/(q+1)}}} \cdot \sigma_t^{q/(q+1)}$$

$$\leq 4 \log K + 7 \sqrt{\sum_{t=1}^{T} \sigma_t^{q/(q+1)} \log K},$$

where the last inequality follows from

$$\sum_{i=1}^{n} \frac{a_i}{\sqrt{\sum_{j \le i} a_j}} \le \sum_{i=1}^{n} \int_{\sum_{j \le i-1} a_j}^{\sum_{j \le i} a_j} \frac{\mathrm{d}x}{\sqrt{x}} = \int_{0}^{\sum_{i=1}^{n} a_i} \frac{\mathrm{d}x}{\sqrt{x}} = 2\sqrt{\sum_{i=1}^{n} a_i}$$

for any non-negative reals $a_1, \cdots, a_n$. Plugging the above upper bound of $\mathbb{E}[V_T]$ into (9) completes the proof of the lemma. □

Theorem 1 follows from Lemma 1 and the following Jensen's inequality:

$$\sqrt{\log T \cdot \sum_{t=1}^{T} \sigma_t^{\frac{q}{q+1}}} \le \sqrt{\log T \cdot T \cdot \left( \frac{\sum_{t=1}^{T} \sigma_t}{T} \right)^{\frac{q}{q+1}}} \le \sqrt{\log T \cdot L^{q/(q+1)} \cdot T^{1/(q+1)}}.$$

### B.1.2  Proof of Upper Bound in Theorem 2.

To achieve the upper bound of Theorem 2, we construct the same $T$ base experts as Algorithm 1, as well as $T$ additional experts who bid $h_t + i/T, i \in [T]$ at each time $t$. Then at an additional $O(1)$ cost in the final regret, the additional experts include an expert who bids $h_t + \sqrt{L/T}$ at each time $t$. Using the same analysis in the proof of Lemma 1, this algorithm achieves a regret upper bound

$$\mathrm{Reg}(\pi_2) \le 2 \left( \frac{\log(2T)}{\eta} + \eta \cdot \sum_{t=1}^{T} \mathbb{E}[r_t^*] \right),$$

where $\eta > 0$ is a fixed learning rate, and

$$r_t^* \le \begin{cases} h_t + \sqrt{L/T} - m_t & \text{if } h_t + \sqrt{L/T} \ge m_t \\ 1 & \text{if } h_t + \sqrt{L/T} < m_t \end{cases}.$$

Consequently,

$$\sum_{t=1}^{T} \mathbb{E}[r_t^*] \le \sum_{t=1}^{T} \mathbb{E}[|h_t + \sqrt{L/T} - m_t|] + \sum_{t=1}^{T} \mathbb{P}(h_t + \sqrt{L/T} < m_t)$$

$$\le \sqrt{LT} + \mathbb{E}\left[ \sum_{t=1}^{T} |h_t - m_t| \right] + \frac{1}{\sqrt{L/T}} \mathbb{E}\left[ \sum_{t=1}^{T} |h_t - m_t| \right]$$

$$\le 2\sqrt{LT} + L \le 3\sqrt{LT},$$

as $1 \le L \le T$. Now choosing $\eta = \min\{1/4, \sqrt{(\log T)/\sqrt{LT}}\}$ leads to the regret upper bound $O\left( (\log T)^{\frac{1}{2}} (T \cdot L)^{\frac{1}{4}} \right)$.

## B.2  Proof of Regret Lower Bounds in Theorem 1 and Theorem 2

### B.2.1  Proof of Lower Bound in Theorem 1.

*Proof.* We use Le Cam's Two-Point method. Construct hint and minimum bid to win as follows: Let $h_t = \frac{1}{2}, t = 1, \ldots, T$ and $\sigma_t$ be the same for all $t$ such that $\sigma^{\frac{q}{q+1}} \le \frac{1}{4}$. Consider the following two CDFs for $m_t \in [0, 1]$:

$$G_1(x) = \begin{cases} 0, & \text{if } 0 < x < \frac{1}{2} \\ 2 \cdot (1 - \bar{x} + \delta), & \text{if } \frac{1}{2} < x < \bar{x} \\ 1, & \text{if } \bar{x} < x < 1 \end{cases}, \quad G_2(x) = \begin{cases} 0, & \text{if } 0 < x < \frac{1}{2} \\ 2 \cdot (1 - \bar{x} - \delta), & \text{if } \frac{1}{2} < x < \bar{x} \\ 1, & \text{if } \bar{x} < x < 1 \end{cases},$$

where $\bar{x} := \frac{1}{2} + \frac{1}{2} \cdot \sigma^{\frac{q}{q+1}}$ and let $\delta < \frac{1}{2} \cdot \sigma^{\frac{q}{q+1}}$. Easy to observe the above construction satisfies:

$$\mathbb{E}[|m_t - h_t|^q] \leq 2 \cdot (\frac{1}{2} \cdot \sigma^{\frac{q}{q+1}} + \delta) \cdot \left(\frac{1}{2} \cdot \sigma^{\frac{q}{q+1}}\right)^q \leq \sigma^{\frac{q}{q+1}} \cdot \left(\sigma^{\frac{q}{q+1}}\right)^q = \sigma^q.$$

Let $r_1(v_t, b_t)$ and $r_2(v_t, b_t)$ be the expected instantaneous reward under CDFs $G_1$ and $G_2$. Then under the above construction:

$$\max_{b \in [0,1]} r_1(1, b) = r_1(1, \frac{1}{2}) = \frac{1}{2} \cdot \frac{1 - \bar{x} + \delta}{1 - \frac{1}{2}} = 1 - \bar{x} + \delta,$$

$$\max_{b \in [0,1]} r_2(1, b) = r_2(1, \bar{x}) = 1 - \bar{x},$$

$$\max_{b \in [0,1]} (r_1(1, b) + r_2(1, b)) = r_1(1, \bar{x}) + r_2(1, \bar{x}) = 2 \cdot (1 - \bar{x}).$$

Therefore, for any $b_t \in [0, 1]$,

$$\left(\max_{b \in [0,1]} r_1(1, b) - r_1(1, b_t)\right) + \left(\max_{b \in [0,1]} r_2(1, b) - r_2(1, b_t)\right)$$

$$\geq \left(\max_{b \in [0,1]} r_1(1, b)\right) + \left(\max_{b \in [0,1]} r_2(1, b)\right) - \max_{b \in [0,1]} (r_1(1, b) + r_2(1, b))$$

$$= (1 - \bar{x} + \delta) + (1 - \bar{x}) - 2 \cdot (1 - \bar{x}) = \delta.$$

Thus we have for any policy $\pi$,

$$\sup_G \text{Reg}(\pi) \geq \frac{1}{2} \mathbb{E}_{G_1}[\text{Reg}(\pi)] + \frac{1}{2} \mathbb{E}_{G_2}[\text{Reg}(\pi)]$$

$$= \frac{1}{2} \sum_{t=1}^{T} \left( \mathbb{E}_{P_1^t}\left[\max_{b \in [0,1]} r_1(1, b) - r_1(1, b_t)\right] + \mathbb{E}_{P_2^t}\left[\max_{b \in [0,1]} r_2(1, b) - r_2(1, b_t)\right]\right) \tag{10}$$

$$\geq \frac{1}{2} \sum_{t=1}^{T} \delta \cdot \int \min\{dP_1^t, dP_2^t\}$$

$$\geq \frac{1}{2} \sum_{t=1}^{T} \delta \cdot \left(1 - \|P_1^t - P_2^t\|_{\text{TV}}\right)$$

$$\geq \frac{1}{2} T \delta \cdot \left(1 - \|P_1^T - P_2^T\|_{\text{TV}}\right),$$

where $b_t$ in (10) denotes the bid of the oracle chosen by policy $\pi$ at time $t$ and $P_i^t$ ($i \in \{1, 2\}$) denotes the distribution of all observables $(m_1, \ldots, m_{t-1})$ at the beginning of time $t$. The KL divergence:

$$D_{\text{KL}}(P_1^T \| P_2^T) = (T - 1) \cdot D_{\text{KL}}(G_1 \| G_2)$$

$$= (T - 1) \cdot \left(2 \cdot (1 - \bar{x} + \delta) \cdot \log \frac{1 - \bar{x} + \delta}{1 - \bar{x} - \delta} + 2 \cdot \left(\bar{x} - \frac{1}{2} - \delta\right) \cdot \log \frac{\bar{x} - \frac{1}{2} - \delta}{\bar{x} - \frac{1}{2} + \delta}\right)$$

$$\leq (T - 1) \cdot \left(2 \cdot (1 - \bar{x} + \delta) \cdot \left(\frac{1 - \bar{x} + \delta}{1 - \bar{x} - \delta} - 1\right) + 2 \cdot \left(\bar{x} - \frac{1}{2} - \delta\right) \cdot \left(\frac{\bar{x} - \frac{1}{2} - \delta}{\bar{x} - \frac{1}{2} + \delta} - 1\right)\right)$$

$$= 4 \cdot \delta \cdot (T - 1) \cdot \left(\frac{1 - \bar{x} + \delta}{1 - \bar{x} - \delta} - \frac{\bar{x} - \frac{1}{2} - \delta}{\bar{x} - \frac{1}{2} + \delta}\right)$$

$$\leq \frac{4T \cdot \delta^2}{(\bar{x} - \frac{1}{2} + \delta)(1 - \bar{x} - \delta)}$$

$$\leq \frac{16T \cdot \delta^2}{\frac{1}{2} \cdot \sigma^{\frac{q}{q+1}} + \delta}.$$

$$\leq \frac{32T \cdot \delta^2}{\sigma^{\frac{q}{q+1}}}.$$

Taking the separation parameter $\delta = \min\left\{\frac{1}{2} \cdot \sigma^{\frac{q}{q+1}}, \frac{1}{8} \cdot \sigma^{\frac{q}{2(q+1)}} \cdot T^{-\frac{1}{2}}\right\}$ and substituting into (6) leads to the regret lower bound in Theorem 1:

$$\Omega\left(\sqrt{T\sigma^{\frac{q}{q+1}}}\right) = \Omega\left(\sqrt{L^{\frac{q}{q+1}} \cdot T^{\frac{1}{q+1}}}\right).$$

$\square$

### B.2.2   Proof of Lower Bound in Theorem 2.

*Proof.* At each time $t$, let $v_t = 1$ and point estimation equals to $\frac{1}{2}$. Define $\varepsilon \in [0, \frac{1}{8}]$ to be some parameter relevant to $L$. Consider the following two scenarios: (each with probability $\frac{1}{2}$)

- $\sigma_t$ equals to 0 with probability $p_1 := 1 - 2(\varepsilon - \delta)$, and equals to $\varepsilon$ with probability $1 - p_1$, in which case $m_t$ always takes value $h_t + \varepsilon$.

- $\sigma_t$ equals to 0 with probability $p_2 := 1 - 2(\varepsilon + \delta)$, and equals to $\varepsilon$ with probability $1 - p_2$, in which case $m_t$ always takes value $h_t + \varepsilon$.

Easy to observe under this construction the expected value of $L$:

$$\bar{L} = \sum_{t=1}^{T} \frac{\varepsilon}{2} \cdot (2(\varepsilon + \delta) + 2(\varepsilon - \delta)) = 2\varepsilon^2 \cdot T.$$

The above construction also satisfies:

$$\max_{b \in [0,1]} R_1(1, b) = R_1\left(1, \frac{1}{2}\right) = \frac{1}{2} - \varepsilon + \delta,$$

$$\max_{b \in [0,1]} R_2(1, b) = R_2\left(1, \frac{1}{2} + \varepsilon\right) = \frac{1}{2} - \varepsilon,$$

$$\max_{b \in [0,1]} (R_1(1, b) + R_2(1, b)) = R_1\left(1, \frac{1}{2} + \varepsilon\right) + R_2\left(1, \frac{1}{2} + \varepsilon\right) = 2 \cdot \left(\frac{1}{2} - \varepsilon\right),$$

where $R_1$ and $R_2$ are expected rewards under the two scenarios. The following steps are similar to previous subsection, for any policy $\pi$,

$$\sup_{\{m_t, h_t, \sigma_t\}} \text{Reg}(\pi) \geq \frac{1}{2}\mathbb{E}_1[\text{Reg}(\pi)] + \frac{1}{2}\mathbb{E}_2[\text{Reg}(\pi)]$$

$$= \frac{1}{2}\sum_{t=1}^{T}\left(\mathbb{E}_{P_1^t}\left[\max_{b \in [0,1]} R_1(1, b) - R_1(1, b_t)\right] + \frac{1}{2}\mathbb{E}_{P_2^t}\left[\max_{b \in [0,1]} R_2(1, b) - R_2(1, b_t)\right]\right)$$

$$\geq \frac{1}{2}\sum_{t=1}^{T}\delta \cdot \int \min\{dP_1^t, dP_2^t\}$$

$$\geq \frac{1}{2}\sum_{t=1}^{T}\delta \cdot \left(1 - \|P_1^t - P_2^t\|_{\text{TV}}\right)$$

$$\geq \frac{1}{2}T\delta \cdot \left(1 - \|P_1^T - P_2^T\|_{\text{TV}}\right), \tag{11}$$

with $P_1^t$ and $P_2^t$ defined the same as (10). And the KL divergence

$$D_{\text{KL}}(P_1^T \| P_2^T) = \sum_{t=1}^{T} \left( 2(\varepsilon - \delta) \cdot \log \frac{\varepsilon - \delta}{\varepsilon + \delta} + (1 - 2(\varepsilon - \delta)) \cdot \log \frac{1 - 2(\varepsilon - \delta)}{1 - 2(\varepsilon + \delta)} \right)$$

$$\leq \sum_{t=1}^{T} \left( 2(\varepsilon - \delta) \cdot \frac{-2\delta}{\varepsilon + \delta} + (1 - 2(\varepsilon - \delta)) \cdot \frac{4\delta}{1 - 2(\varepsilon + \delta)} \right)$$

$$\leq 4\delta T \cdot \left( -\frac{\varepsilon - \delta}{\varepsilon + \delta} + \frac{1 - 2\varepsilon + 2\delta}{1 - 2\varepsilon - 2\delta} \right)$$

$$= 8\delta^2 T \cdot \frac{1}{(\varepsilon + \delta)(1 - 2\varepsilon - 2\delta)}$$

$$\leq \frac{16T \cdot \delta^2}{\varepsilon}.$$

Taking $\delta = \min\left\{ \varepsilon, \frac{1}{4}\sqrt{\frac{\varepsilon}{2T}} \right\}$ and substitute in (11), we have:

$$\sup_{\{m_t, h_t, \sigma_t\}} \text{Reg}(\pi) \geq \frac{1}{4} \min\left\{ \varepsilon T, \frac{1}{4\sqrt{2}}\sqrt{T \cdot \varepsilon} \right\},$$

which leads to a lower bound of $\Omega((T \cdot L)^{\frac{1}{4}})$. Note that the construction above requires $\sigma_t$ to be unknown, otherwise one can achieve 0 regret by bidding hint for $\sigma_t = 0$ and bidding hint $+ \varepsilon$ for $\sigma = \varepsilon$, which is a technical explanation for the separation in Section 3. $\qquad \square$

### B.3 Proof of Theorem 3.

*Proof.* If $L > \left(\sqrt{T}\right)^{\frac{q-1}{q}}$, then $T^{\frac{1}{q+1}} L^{\frac{q}{q+1}} > \sqrt{T}$ and the regret can be lower bounded by $\Omega\left(\sqrt{T}\right)$. So in the following construction, we assume $L \leq \left(\sqrt{T}\right)^{\frac{q-1}{q}}$. First we divide time horizon to $\left\lfloor T^{\frac{1}{q+1}} L^{\frac{q}{q+1}} \right\rfloor$ equal parts and let $\sigma_t$ be the same for all $t$. Construct private values and hints as follows: For $t = i \cdot \left\lfloor \left(\frac{T}{L}\right)^{\frac{q}{q+1}} \right\rfloor + 1, i \cdot \left\lfloor \left(\frac{T}{L}\right)^{\frac{q}{q+1}} \right\rfloor + 2, \ldots, (i+1) \cdot \left\lfloor \left(\frac{T}{L}\right)^{\frac{q}{q+1}} \right\rfloor$,

$$v_t = \frac{1}{2} + \frac{1}{2} \cdot \frac{i}{T^{\frac{1}{q+1}} L^{\frac{q}{q+1}}},$$

$$h_t = \frac{1}{4} + \frac{i}{4} \cdot \sigma^{\frac{q}{q+1}},$$

$$m_t = \begin{cases} \frac{1}{4} + \frac{i}{4} \cdot \sigma^{\frac{q}{q+1}}, & w.p. \quad 1 - \frac{1}{4} \cdot \left( \sigma^{\frac{q}{q+1}} \pm \delta \right) \\ \frac{1}{4} + \frac{i+1}{4} \cdot \sigma^{\frac{q}{q+1}}, & w.p. \quad \frac{1}{4} \cdot \left( \sigma^{\frac{q}{q+1}} \pm \delta \right) \end{cases}$$

where $i = 0, 1, 2, \ldots, \left\lfloor T^{\frac{1}{q+1}} L^{\frac{q}{q+1}} \right\rfloor - 1$ and $\delta$ is the separation parameter similarly defined in the proof of Theorem 1. Since $L \leq \left(\sqrt{T}\right)^{\frac{q-1}{q}}$, we have

$$\frac{1}{T^{\frac{1}{q+1}} L^{\frac{q}{q+1}}} \geq \left(\frac{L}{T}\right)^{\frac{q}{q+1}} = \sigma^{\frac{q}{q+1}},$$

which ensures any strategy $\pi$ that bids in $\left[ \frac{1}{4} + \frac{i}{4} \cdot \sigma^{\frac{q}{q+1}}, \frac{1}{4} + \frac{i+1}{4} \cdot \sigma^{\frac{q}{q+1}} \right]$ for the $i$-th part belongs to 1-Lipschitz and monotone oracle. Therefore, we can now consider the whole time horizon as $\left\lfloor T^{\frac{1}{q+1}} L^{\frac{q}{q+1}} \right\rfloor$ independent problems, each of which consists of $\left\lfloor \left(\frac{T}{L}\right)^{\frac{q}{q+1}} \right\rfloor$ time steps and has fixed $v_t$. Substituting $L_i := \left\lfloor \left(\frac{T}{L}\right)^{\frac{q}{q+1}} \right\rfloor \cdot \sigma$, which is $L$ for the $i$-th subproblem, and applying similar method

to the proof of Theorem 1, we can get:

$$\sup_G \text{Reg}_i(\pi) = \Omega\left(\sqrt{\left(\frac{T}{\left\lceil T^{\frac{1}{q+1}} L^{\frac{q}{q+1}} \right\rceil}\right)^{\frac{1}{q+1}} \cdot \left(\frac{L}{\left\lceil T^{\frac{1}{q+1}} L^{\frac{q}{q+1}} \right\rceil}\right)^{\frac{q}{q+1}}}\right)$$

$$= \Omega\left(\frac{T^{\frac{1}{q+1}} L^{\frac{q}{q+1}}}{\left\lceil T^{\frac{1}{q+1}} L^{\frac{q}{q+1}} \right\rceil}\right) = \Omega(1),$$

for each independent problem. Summing over all subproblems leads to the lower bound $\Omega\left(T^{\frac{1}{q+1}} L^{\frac{q}{q+1}}\right)$. □

### B.4 Proof of Theorem 4

*Proof.* We prove that even when $L$ takes expected value $\Theta(1)$, the minimax regret is still lower bounded by $\Omega(\sqrt{T})$. The proof is similar to that of Theorem 3, but by dividing time horizon into $\sqrt{T}$ subproblems. At each time $t$ inside the $i$-th subproblem, the bidder observes $h_t = \frac{1}{4} + \frac{i \cdot \varepsilon}{4}$ (where $\varepsilon = \frac{1}{\sqrt{T}}$). In the construction of the lower bound in Theorem 2, $\sigma_t$ equals to 0 with probability $1 - \Theta(\varepsilon)$ and equals to $\varepsilon$ with probability $\Theta(\varepsilon)$. Thus,

$$\bar{L} = \mathbb{E}\left[\sum_{t=1}^{T} \sigma_t\right] = T \cdot \varepsilon^2 = \Theta(1).$$

Meanwhile, applying similar method to the proof of Theorem 2, we can get a lower bound of $\Omega\left(\sqrt{\sqrt{T} \cdot \frac{1}{\sqrt{T}}}\right) = \Omega(1)$ for each independent problem, leading to the final lower bound $\Omega\left(\sqrt{T}\right)$. □

## C Proof of Main Result in Section 4

### C.1 Proof of Theorem 5

#### C.1.1 Proof of Upper Bounds in Theorem 5

*Proof.* In the following subsection, we provide a way to achieve $O\left(\sqrt{\log T \cdot T^{\frac{1}{q+1}} \cdot L^{\frac{q}{q+1}} \cdot K}\right)$ regret upper bound. [*]

Figure 3 shows any function in oracle can be mapped to a piecewise constant function whose value only takes those in the support set, define this mapped function set to be $A$. We prove in the appendix that the number of functions in the converted set A is smaller than $T^K$, then applying the algorithm in Theorem 1's proof directly leads to an upper bound of [*]

$$O\left(\sqrt{\log\left(|\text{expert set}|\right) \cdot T^{\frac{1}{q+1}} \cdot L^{\frac{q}{q+1}}}\right) = O\left(\sqrt{\log T \cdot T^{\frac{1}{q+1}} \cdot L^{\frac{q}{q+1}} \cdot K}\right).$$

To show set $A$ is small enough, let's first imagine walking from $(1, 0)$ to $(T, K)$ with each step either to the positive direction of $x$-axis or $y$-axis exactly by 1. There are $T + K - 1$ steps in total and one may choose $K$ of them to go up. Now given any function in $A$, suppose at $x = 1$ the value equals to the $i$-th support and at $x = T$ the value equals to the $j$-th support, which can be considered as points $(1, i)$ and $(T, j)$, $i, j \in \mathbb{Z}$, $0 \leq i \leq j \leq K$. Without loss of monotonicity, we add points $(0, 0)$ and $(T + 1, K)$ to the interval-support pairs of this function, i.e. the function takes value of the

---

[*]The other two are described in Appendix A.

[*]Although in the proof of Theorem 1 we show an upper bound of $O\left(\sqrt{\log T \cdot T^{\frac{1}{q+1}} \cdot L^{\frac{q}{q+1}}}\right)$, the proof schetch can indeed be applied to any finite set of experts.

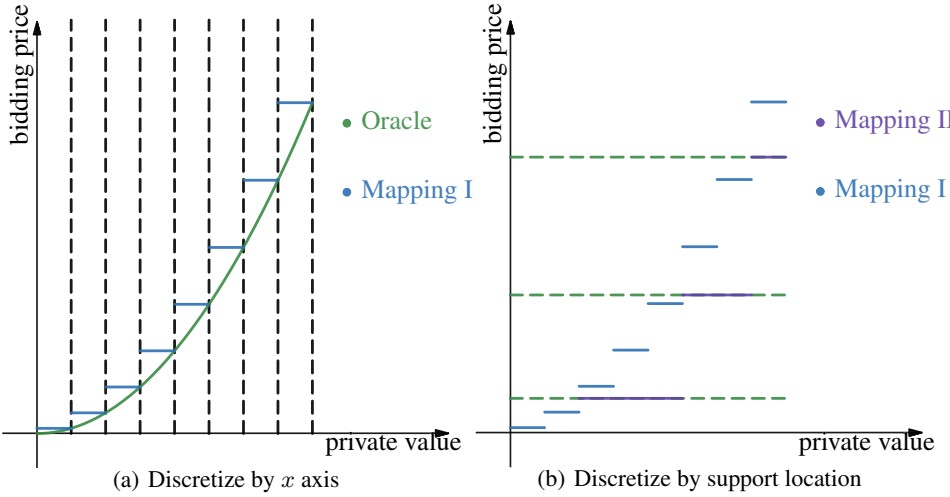

(a) Discretize by $x$ axis    (b) Discretize by support location

Figure 3: Given any 1-Lipschitz and monotone oracle, we first discretize the $x$-axis into $T$ small intervals, changing the oracle to a piecewise constant function that bids the maximum point for each interval in the oracle; Secondly, we map this piecewise constant function to a piecewise function that only takes support value as bidding price. Easy to verify step 1 leads to $T \cdot O(\frac{1}{T}) = O(1)$ loss, while step 2 leads to a non-negative change to the cumulative reward.

$i$-th support for the $t$-th interval, $i \in [K]$, $t \in [T]$, iff we pass point $(t, i)$ in the route from $(0, 0)$ to $(T + 1, K)$. The set of routes and set $A$ forms a bijection, both have cardinality:

$$\binom{T + K - 1}{K} = \frac{T + K - 1}{K} \cdot \frac{T + K - 2}{K - 1} \cdots \frac{T}{1} \leq T^K.$$

$\square$

### C.1.2  Proof of Lower Bounds in Theorem 5

*Proof.* Consider the three cases separately:

- If $L < \frac{K^{\frac{q+1}{q}}}{T^{\frac{1}{q}}}$, then as in the proof of Theorem 3 we can construct $N = \left\lfloor T^{\frac{1}{q+1}} L^{\frac{q}{q+1}} \right\rfloor$ independent problems since $N < K$ in this case. For each independent problem the lower bound is $\Omega(1)$, leading to a total lower bound of $\Omega\left(T^{\frac{1}{q+1}} \cdot L^{\frac{q}{q+1}}\right)$.

- If $\frac{K^{\frac{q+1}{q}}}{T^{\frac{1}{q}}} \leq L \leq \frac{T}{K^{\frac{q+1}{q}}}$, we cannot divide into $\left\lfloor T^{\frac{1}{q+1}} L^{\frac{q}{q+1}} \right\rfloor$ subproblems since there are only $K$ values $m_t$ can take. So instead, we divide time horizon into $K$ subproblems:
  For $t = i \cdot \lfloor \frac{T}{K} \rfloor + 1, i \cdot \lfloor \frac{T}{K} \rfloor + 2, \ldots, (i + 1) \cdot \lfloor \frac{T}{K} \rfloor$,
  $$v_t = \frac{1}{2} + \frac{1}{2} \cdot \frac{i}{K},$$
  $$h_t = \frac{1}{4} + \frac{i}{4} \cdot \sigma^{\frac{q}{q+1}},$$
  $$m_t = \begin{cases} \frac{1}{4} + \frac{i}{4} \cdot \sigma^{\frac{q}{q+1}}, & w.p. \quad 1 - \frac{1}{4} \cdot \left(\sigma^{\frac{q}{q+1}} \pm \delta\right) \\ \frac{1}{4} + \frac{i+1}{4} \cdot \sigma^{\frac{q}{q+1}}, & w.p. \quad \frac{1}{4} \cdot \left(\sigma^{\frac{q}{q+1}} \pm \delta\right) \end{cases}$$
  where $i = 0, 1, 2, \ldots, K-1$. Observe that the difference between $v_t$ for adjacent subproblem is $\frac{1}{2} \cdot \frac{1}{K}$ and the difference between bid value for adjacent subproblem is at most
  $$2 \cdot \frac{\sigma^{\frac{q}{q+1}}}{4} = \frac{\sigma^{\frac{q}{q+1}}}{2} = \frac{L^{\frac{q}{q+1}}}{2 \cdot T^{\frac{q}{q+1}}} \leq \frac{1}{2} \cdot \frac{1}{K},$$

ensuring the $N = K$ subproblems are indeed independent from each other. Additionally, the separation parameter $\delta$ for each subproblem equals to $\sqrt{\frac{\sigma^{\frac{q}{q+1}}}{\frac{T}{K}}} = \sqrt{\frac{K \cdot L^{\frac{q}{q+1}}}{T^{\frac{1}{q+1}}}}$, which is smaller than the separation of $m_t$: $\sigma^{\frac{q}{q+1}} = \left(\frac{L}{T}\right)^{\frac{q}{q+1}}$. Thus substituting Theorem 1, finally the lower bound is,

$$K \cdot \Omega\left(\left(\frac{T}{K}\right)^{\frac{1}{q+1}} \cdot \left(\frac{L}{K}\right)^{\frac{q}{q+1}}\right) = \Omega\left(\sqrt{K \cdot T^{\frac{1}{q+1}} \cdot L^{\frac{q}{q+1}}}\right).$$

- If $L > \frac{T}{K^{\frac{q+1}{q}}}$, a traditional lower bound gives $\Omega\left(\sqrt{T}\right)$.

$\square$

## C.2 Proof of Theorem 6

*Proof.* Let the learning rate for the upper level $\eta_{t,2} = \min\left\{\frac{1}{4}, \sqrt{\frac{\log 3}{\left\lfloor \sum_{s=1}^{t-1} \sigma_s^{\frac{q}{q+1}} \right\rfloor + 1}}\right\}$ and apply similar analysis as in Appendix B.1.1:

$$\sum_{t=1}^{T} \mathbb{E}[X_t] \geq \max_{i \in \{f,g,h\}} \sum_{t=1}^{T} r_{t,i} - 2 \cdot \left(\frac{\log 3}{\eta_{T,2}} + 2\sum_{t=1}^{T} \eta_{t,2} \cdot 2 \cdot \sigma_t^{\frac{q}{q+1}}\right)$$

$$= \max_{i \in \{f,g,h\}} \sum_{t=1}^{T} r_{t,i} - 2 \cdot \left(\sqrt{\log 3 \cdot \sum_{t=1}^{T} \sigma_t^{\frac{q}{q+1}}} + 4\sqrt{\log 3} \cdot \sum_{t=1}^{T} \frac{\sigma_t^{\frac{q}{q+1}}}{\sqrt{\left\lfloor \sum_{s=1}^{t} \sigma_s^{\frac{q}{q+1}} \right\rfloor + 1}}\right)$$

$$\overset{(a)}{\geq} \max_{i \in \{f,g,h\}} \sum_{t=1}^{T} r_{t,i} - 2 \cdot \left(\sqrt{\log 3 \cdot \sum_{t=1}^{T} \sigma_t^{\frac{q}{q+1}}} + 8\sqrt{\log 3 \cdot \sum_{t=1}^{T} \sigma_t^{\frac{q}{q+1}}}\right)$$

$$= \max_{i \in \{f,g,h\}} \sum_{t=1}^{T} r_{t,i} - 18 \cdot \sqrt{\log 3 \cdot \sum_{t=1}^{T} \sigma_t^{\frac{q}{q+1}}}, \tag{12}$$

where $\sum_{t=1}^{T} \mathbb{E}[X_t]$ is the expected total reward by running Algorithm 1, with expectation taken over both policy randomness and possible $m_t$ sequences. (a) can be considered as taking integral of function $f(x) = \frac{1}{\sqrt{x}}$, but with another piecewise function smaller than it instead. And applying similar method to the lower level of the first node we have:

$$\sum_{t=1}^{T} r_{t,f} \geq \max_{a \in [T^K]} \sum_{t=1}^{T} r_{t,a} - 18 \cdot \sqrt{K \log T \cdot \sum_{t=1}^{T} \sigma_t^{\frac{q}{q+1}}}. \tag{13}$$

Combining (12) and (13) and the regret upper bound of ChEW algorithm and choosing hint expert:

$$\sum_{t=1}^{T} \mathbb{E}[X_t] \geq \max_{i \in \{f,g,h\}} \left(\max_{a \in [T^K]} \sum_{t=1}^{T} r_{t,a} - \text{Reg}(i)\right) - 18 \cdot \sqrt{\log 3 \cdot \sum_{t=1}^{T} \sigma_t^{\frac{q}{q+1}}}$$

$$= \max_{a \in [T^K]} \sum_{t=1}^{T} r_{t,a} - \min\left\{18 \cdot \sqrt{K \log T \cdot \sum_{t=1}^{T} \sigma_t^{\frac{q}{q+1}}}, 2 \cdot \sum_{t=1}^{T} \sigma_t^{\frac{q}{q+1}}, C \cdot \sqrt{T}\right\} - 18 \cdot \sqrt{\log 3 \cdot \sum_{t=1}^{T} \sigma_t^{\frac{q}{q+1}}},$$

where $C$ is a constant number. Therefore, we have:

$$\text{Reg}(\pi) = O\left(\min\left\{\sqrt{\log T \cdot \sum_{t=1}^{T} \sigma_t^{\frac{q}{q+1}} \cdot K}, \sum_{t=1}^{T} \sigma_t^{\frac{q}{q+1}}, \sqrt{T}\right\}\right) + O\left(\sqrt{\log 3 \cdot \sum_{t=1}^{T} \sigma_t^{\frac{q}{q+1}}}\right)$$

$$\overset{(b)}{=} O\left(\min\left\{\sqrt{\log T \cdot \sum_{t=1}^{T} \sigma_t^{\frac{q}{q+1}} \cdot K}, \sum_{t=1}^{T} \sigma_t^{\frac{q}{q+1}}, \sqrt{T}\right\}\right),$$

while (b) holds since $\sum_{t=1}^{T} \sigma_t^{\frac{q}{q+1}} > L > 1$. $\qquad\square$

### C.3 Proof of Theorem 7

#### C.3.1 Proof of Upper Bound in Theorem 7

*Proof.* Instead of one single hint expert in Algorithm 2, construct $T$ hint experts, with each one bidding a constant gap over $h_t$, i.e. with the first hint expert bidding $h_t + \frac{1}{T}$ for $t = 1, \ldots, T$; the second hint expert bidding $h_t + \frac{2}{T}$ for $t = 1, \ldots, T$; etc. The upper layer then consists of $T$ hint experts and two super nodes, representing ChEW algorithm ($g$) and modified Algorithm 1 ($f$). The lower layer of $f$ consists of $T^K$ base experts (constructed as in Appendix C.1) and $T$ hint experts. Let the learning rate for the upper level $\eta_2 = \min\left\{\frac{1}{4}, \sqrt{\frac{\log(T+2)}{\sqrt{TL}}}\right\}$,

$$\sum_{t=1}^{T} \mathbb{E}[X_t] \geq \max_{i \in \{f,g,h\}} \sum_{t=1}^{T} r_{t,i} - 2 \cdot \left(\frac{\log(T+2)}{\eta_2} + 4\eta_2\sqrt{LT}\right)$$

$$= \max_{i \in \{f,g,h\}} \sum_{t=1}^{T} r_{t,i} - 10 \cdot \sqrt{\log(T+2)} \cdot \sqrt{LT}, \qquad (14)$$

And applying similar method to super node $f$:

$$\sum_{t=1}^{T} r_{t,f} \geq \max_{a \in [T^K]} \sum_{t=1}^{T} r_{t,a} - 10 \cdot \sqrt{K \log T \cdot \sqrt{TL}}. \qquad (15)$$

Combining (14) and (15),

$$\sum_{t=1}^{T} \mathbb{E}[X_t] \geq \max_{i \in \{f,g,h\}} \left(\max_{a \in [T^K]} \sum_{t=1}^{T} r_{t,a} - \text{Reg}(i)\right) - 20 \cdot \sqrt{\log T \cdot \sqrt{TL}}$$

$$= \max_{a \in [T^K]} \sum_{t=1}^{T} r_{t,a} - \min\left\{10 \cdot \sqrt{K \log T \cdot \sqrt{TL}}, 2 \cdot \sqrt{TL}, C \cdot \sqrt{T}\right\} - 20 \cdot \sqrt{\log T \cdot \sqrt{TL}},$$

where $C$ is a constant number. Therefore, we have:

$$\text{Reg}(\pi) = O\left(\min\left\{\sqrt{K \log T \cdot \sqrt{TL}}, \sqrt{T}\right\}\right) + O\left(\sqrt{\log T \cdot \sqrt{TL}}\right)$$

$$= O\left(\min\left\{\sqrt{K \log T \cdot \sqrt{TL}}, \sqrt{T \log T}\right\}\right).$$

$\qquad\square$

#### C.3.2 Proof of Lower Bound in Theorem 7

The following is similar to proof of lower bound in Theorem 5.

*Proof.* • If $L > \frac{T}{K^2}$, as in the proof of Theorem 4 construct $N_0 = \left\lfloor\sqrt{\frac{T}{L}}\right\rfloor < K$ independent sub-problems, while for each sub-problem

$$L' = \frac{L}{\sqrt{T/L}} = \sqrt{\frac{L^3}{T}}, \quad T' = \frac{T}{\sqrt{T/L}} = \sqrt{TL},$$

and for each sub-problem regret is lower bounded by $\Omega\left(\left(\sqrt{LT} \cdot \sqrt{\frac{L^3}{T}}\right)^{1/4}\right)$, leading to

a total lower bound of $\Omega\left(\left(\sqrt{LT} \cdot \sqrt{\frac{L^3}{T}}\right)^{1/4} \cdot \sqrt{\frac{T}{L}}\right) = \Omega(\sqrt{T})$.

- If $L \leq \frac{T}{K^2}$, it is not feasible to construct $N_0$ independent sub-problems as the optimal bidding value can not take $N_0 > K$ values. Instead construct $K$ independent problems, with the separation parameter (see Appendix B.2.2): $\delta = \sqrt{\frac{L}{T}} \cdot \frac{K}{T} < \frac{1}{T}$, leading to a total regret lower bound of $\Omega\left(\sqrt{\sqrt{\frac{T}{K} \cdot \frac{L}{K}}} \cdot K = \Omega\left(\sqrt{K \cdot \sqrt{TL}}\right)\right)$.

$\square$

# D  Experimental Details

## D.1  Description of Experiment 1 in Section 5

Divide the whole range of private value to $D$ bins, each of which contains $v_t$'s that are close to each other. As long as the bidder observes $v_t$ at time $t$, we reduce the problem to the bin focusing on the data points with private values close to $v_t$. Then each bin itself forms a sub-problem described in Section 3. Experiment 1 only serves as an illustration of the effect by hints. The role of hints is threefolds:

- We use hint to help allocating data to different bins. Instead of binning only by private values, we use hint as a side information and conduct binning also based on it. The total number of bins is $M_1 \cdot M_2$, while $M_1$ is the number of discretization for $v_t$ and $M_2$ is the number of discretization for hints. As for the result on empirical data, we observe $M_2 = 4$ already leads to rather good performance.

- We use hint to calculate the estimation of instantaneous reward for any given bid $b'_t$ under the assumption that $m_t = b_t$: $r'_{t,a} := r(b_t; h_t, v_t)$, where $b_t$ is the bid at time $t$ according to oracle $a$. Then we add this estimated reward to each experts' reward history while sampling among these experts:

$$p_{t,a} = \frac{\exp\left(\eta_t \cdot \left(\sum_{s=1}^{t-1} r_{s,a} + r'_{t,a}\right)\right)}{\sum_{a' \in \mathcal{F}} \exp\left(\eta_t \cdot \left(\sum_{s=1}^{t-1} r_{s,a'} + r'_{t,a'}\right)\right)}, t = 2, 3, \cdots, T.$$

And if $\sigma_t$ is also observed, we define $r'_{t,a} := r(b_t; h_t + c_1 \cdot \sigma_t, v_t)$ instead, where $c_1$ is a hyper-parameter to be tuned.

- We include a set of hint experts

$$b_t(a_i) := h_t + \sigma_t^{\Delta_i}, \quad i = 1, 2, \cdots, k,$$

which is close to a combination of algorithms for whether knowing the error, since for real datasets $q$ is often not observed.

The results in Figure 4 shows the improvement by incorporating hint on other two datasets. The results implies that on datasets whose hint has rather small error, e.g. on dataset 1 bidding hint itself already beats simple online learning algorithm, the improvement by hint is more significant. Namely, 4.38% on dataset 1 with more accurate hint and 3.54% on dataset 2 whose hint is not so good.

## D.2  Polynomial Algorithm in Section 5

Consider any 1-Lipschitz & monotone oracle $f$, since support size is finite, $f$ can be mapped to a discontinuous function $f'$ with $O(1)$ loss, which can be further represented by a series of interval-support pair:

$$\left(0, \frac{1}{D}\right] \leftrightarrow s_{i_1}, \quad \left(\frac{1}{D}, \frac{2}{D}\right] \leftrightarrow s_{i_2}, \quad \ldots, \quad \left(\frac{D-1}{D}, 1\right] \leftrightarrow s_{i_D},$$

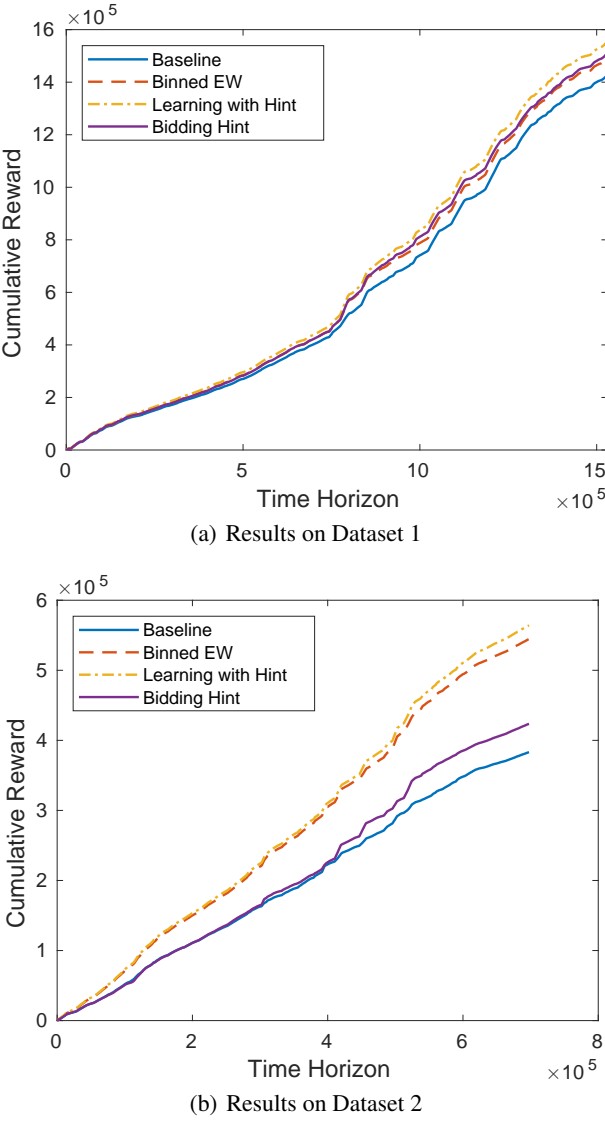

(a) Results on Dataset 1

(b) Results on Dataset 2

Figure 4: Cumulative rewards as a function of time. The dashdot lines stands for incorporating hint into exponential weighting, and the purple solid lines are directly bidding hint. The dotted lines represent binned exponential algorithm.

**Algorithm 3:** DP algorithm without knowing support locations

---

**Inputs:** Time horizon $T$; support size $K$;
**Initialization:** $\text{Reward}_{T,K,T} \leftarrow 0$; $P \leftarrow 0$;
**for** $t = 1, 2, \ldots, T$ **do**

    *% Calculate Sum_Forward&Sum_Backward Matrix*
    $\text{Sum\_Forward}_{T,K,T} \leftarrow 1$; $\text{Sum\_Backward}_{T,K,T} \leftarrow 1$;
    **for** $i = 1, 2, \ldots T$ **do**

        **for** $j = 1, 2, \ldots, T$ **do**

            $\text{Sum\_Forward}_{i,1,j} \leftarrow \text{Sum\_Forward}_{i-1,1,j} \cdot \exp(\eta_t \cdot \text{Reward}_{i,1,j})$;
            $\text{Sum\_Backward}_{i,K,j} \leftarrow \text{Sum\_Backward}_{i+1,K,j} \cdot \exp(\eta_t \cdot \text{Reward}_{i,K,j})$;
            **for** $k = 2, 3, \ldots K - 1$ **do**

$$\text{Sum\_Forward}_{i,k,j} \leftarrow \sum_{v=1}^{j-1} \Big( \text{Sum\_Forward}_{i-1,k-1,v} \cdot \exp(\eta_t \cdot \text{Reward}_{i,k,j}) \Big)$$
$$+ \text{Sum\_Forward}_{i-1,k,j} \cdot \exp(\eta_t \cdot \text{Reward}_{i,k,j});$$

$$\text{Sum\_Backward}_{i,k,j} \leftarrow \sum_{v=j+1}^{T} \Big( \text{Sum\_Backward}_{i+1,k+1,v} \cdot \exp(\eta_t \cdot \text{Reward}_{i,k,j}) \Big)$$
$$+ \text{Sum\_Backward}_{i+1,k,j} \cdot \exp(\eta_t \cdot \text{Reward}_{i,k,j});$$

            **end**

        **end**

    **end**
    *% Calculate Probability*
    $i \leftarrow \lfloor v_t \cdot T \rfloor$;
    **for** $j = 1, 2, \ldots T$ **do**

$$P_j \leftarrow \sum_{k=1,2,\ldots K} \Bigg( \Big( \text{Sum\_Forward}_{i-1,k,j} + \sum_{v=1}^{j-1} \text{Sum\_Forward}_{i-1,k-1,v} \Big) \cdot \exp(\eta_t \cdot \text{Reward}_{i,k,j})$$
$$\cdot \Big( \text{Sum\_Backward}_{i+1,k,j} + \sum_{v=j+1}^{T} \text{Sum\_Backward}_{i+1,k+1,v} \Big) \Bigg);$$

    **end**
    **for** $k = 1, 2, \ldots K$ **do**
        $P_{\lfloor h_t \cdot T \rfloor} \leftarrow P_{\lfloor h_t \cdot T \rfloor} + \exp(\eta_t \cdot \text{RH})$;
    **end**
    Sample $b_t \sim (P / \sum(P))$;
    *% Update Reward Matrix*
    **for** $k = 1, 2, \ldots, K$ **do**
        **for** $j = 1, 2, \ldots T$ **do**
            **if** $m_t \leq j/T$ **then**
                $\text{Reward}_{i,k,j} \leftarrow \text{Reward}_{i,k,j} + (v_t - j/T)$;
        **end**
    **end**
    $\text{RH} \leftarrow \text{RH} + r(h_t; v_t, m_t)$;
**end**

---

where $0 \leq s_1 \leq s_2 \leq s_3 \leq \cdots \leq s_K \leq 1$ are the locations of supports in increasing order and $0 \leq i_1 \leq i_2 \leq \cdots \leq i_D \leq K, i_1, i_2, \cdots, i_D \in \mathbb{Z}$. The main idea is to record the cumulative reward for all possible interval-support tuples and use dynamic programming to calculate total reward for some expert sets instead of keeping track of all $T^K$ experts.

Reward[D][K][D]: The first two dimensions represent interval: $[d][k] : (d/D, (d+1)/D] \leftrightarrow s_k$. The third dimension represent the bidding, with steply update

$$\text{Reward}_{i,k,j} \leftarrow \text{Reward}_{i,k,j} + (v_t - j/D)$$

Then we use dynamic programming to calculate the sum of the rewards for several continuous intervals, instead of keeping track of all $T^K$ experts.

Sum_Forward[D][K][D]: Forward DP recording array, representing combined intervals: $[d][K] : (0, d/D] \leftrightarrow \{1, \ldots, s_k\}$ and the third dimension represents bidding for the last interval: $(d/D, (d+1)/D]$. The update calculation is carried out per step before choosing an action.

Sum_Backward[D][K][D]: Backward DP recording array, representing combined intervals: $[d][K] : ((d+1)/D, 1] \leftrightarrow \{s_k + 1, \ldots, K\}$ and the third dimension represents bidding for the first interval: $((d+1)/D, (d+2)/D]$. The update calculation is carried out per step before choosing an action.

Combining the results of Sum_Forward and Sum_Backward, we can calculate reward history for a subset of the $T^K$ experts, which is the only needed quantity for calculating probability in exponential weighting instead of keeping record with an exponential size.