# OpenReview forum: "Leveraging the Hints: Adaptive Bidding in Repeated First-Price Auctions"
_NeurIPS.cc/2022/Conference — NeurIPS 2022 Accept_

### Official Review · Reviewer_SqbQ · 2022-07-06

**Rating:** 7
**Confidence:** 4
**Soundness:** 4 excellent
**Presentation:** 3 good
**Contribution:** 3 good

**Summary:**

This paper focus on the problem of deriving a best-response bidding strategy in a first-price auction in a bandit setting where a so-called "hint" $h_t$ on the highest bid from the competition (the other bidders) is provided. The paper refines existing upper and lower bound on the regret by parametrizing them with the total error made by the hints, inspired by recent path-length parametrizations of regret in bandit.

The main difference is that, usually, in the literature, a hint $tilde{r}_t(b)$ is provided as an indication of the reward function $r(b, m_t)$ itself. Here the hint $h_t$ informs about the parameter $m_t$ of the reward function.

The paper provides an algorithm with matching upper and lower bounds in 3 settings about 2 types of hints.
Types of hints: just a prediction of $m_t$ or an interval that contains $m_t$ with high proba.
Types settings:
* Constant value $v_t :=1$ and $m_t\in [0,1]$
* Varying values $v_t \in [0,1]$ and $m_t\in[0,1]$
* Varying values $v_t \in [0,1]$ and $m_t$ belonging to a finite subset of $[0,1]$

**Questions:**

1. With the same definition for the hint $h_t$, would it be possible / relevant to provide regret bounds parametrized in terms of cumulative reward prediction error $\tilde{L} = \sum_{t=1}^T \int \big| (v_t - b) \big(\mathbb{1}[b\geq m_t] - \mathbb{1}[b\geq h_t]\big) \big| {\rm d}b$ rather than cumulative competition prediction error $L = \sum_{t=1}^T \sigma_t$ ?

2. Following the previous question, may it provide finer parametrized bound for the case when $v_t$ varies (Th. 3 and 4)? Even though in my understanding of the proof, it won't improve the lower-bound I think.

3. Would the results of the paper extend to best-response to any (nice enough) auction? Like for instance for C-Lipschitz auctions which have a regret without hint of order $CT^{2/3}$ (see [2, Prop. 4.2]).

[2] Learning in repeated auctions, Nedelec et al. (2022)

**Limitations:**

I would have liked to have a bit more motivation on the practical use of hints in auctions. Where does it come from ? Who is providing it ? What data is it based on ?

**Strengths And Weaknesses:**

(+) To my knowledge, the paper does not consists in a simple application of previous technique to the problem of bidding in first-price auction. Indeed, usually, the hint concerns the reward function it self and not a parameter of the reward function. Here, the parametrization of the regret is based on the cumulative prediction error of the parameter and not on the cumulative error of the induced reward function.

(+) I find the result of Th. 3 and 4 unexpected, especially the lower-bound part, even though they feel a bit frustrating.

(-) I would have expected this somehow negative result (of Th. 3 and 4) to be more commented and some intuition as for the why detailed. Especially, this degradation of the regret is introduced as a result of considering varying (rather than constant) values $v_t$, but is solved by considering $m_t$ to have finite support. I completely fail to follow the logic and the intuition.

Minor detail:
* line 48 it is not optimal for an exchange bidding in a header bidder to bid the second highest internal value see [1] for instance
* I think the paper could shorten a lot the introduction. The study of first-price auctions is widespread enough not to need such a long motivation in introduction.

[1] Bidder collusion. Marshall (2007)

---

> ### Author Response · Authors · 2022-08-02
> **Response to reviewer SqbQ**
>
> We thank the reviewer for the thoughtful review and positive feedback. Here we provide our point-to-point response.
>
> The main reason for the negative results (Thm 3 and 4) lies in that private values are continuous (infinite), yet the reward function is discontinuous, thus allowing an adversary to significantly degrade the bidding performance through minor changes of $m_t$ around the bidding price. We mitigated this feedback by introducing finite supports, which limits the adversary's ability to perturb around bidder's bids, and therefore lead to positive results.
>
> Thank you for your suggestion on shortening the introduction - we will implement it in the final version.
>
> Regarding the suggestion of using $\tilde{L}$ instead of $L$, we remark that $\tilde{L}$ leads to the same results as $L$ when $q=\infty$. This is because when $v_t > \max(b_t, h_t)$, integrating over $b\in [\min(b_t, h_t), \max(b_t, h_t)]$ leads to $\tilde{L} = \sum_t (v_t - (b_t+h_t)/2)|b_t - h_t|$. Consequently, $\tilde{L}\le \sum_t |b_t - h_t|$, and as long as $(b_t+h_t)/2$ is not too close to $v_t$ (which holds in our lower bound construction) we have $\tilde{L}\gtrsim \sum_t |b_t-h_t|$. Therefore, in terms of the order of magnitude, both the upper and lower bounds will not change when we move from $L$ to $\tilde{L}$.
>
> Regarding your reference on obtaining an $O(T^{2/3}L^{1/3})$ regret for general Lipschitz auctions (here $L$ is the number of different possible valuations), in fact the main contribution of the literature [HZF+20] was that in first-price auctions, a much smaller regret $\widetilde{O}(\sqrt{T})$ is attainable. First, the exponent of $T$ improves from $2/3$ to $1/2$; second, there is no dependence on $L$ even if $L=\infty$. Also the discretization idea was also applied there, both the algorithm and analysis there are more delicate to arrive at a tight dependence. This is a nice nature of considering a specific auction mechanism (such as first-price auctions) where one could devise an algorithm with a smaller regret, a philosophy we adopted in our work.
>
> On the practical use of hints, we would like to describe our real data in more detail. Our industrial partner could observe some contextual information for the auctions (e.g. browser cookies, ad type, attribute of publisher site, competitor information), from which they fit the contextual information into some model and obtain the context-based prediction $h_t$ (and possibly its standard deviation estimate $\sigma_t$) for the maximum competing bid $m_t$.

---

### Official Review · Reviewer_Bere · 2022-07-12

**Rating:** 6
**Confidence:** 3
**Soundness:** 3 good
**Presentation:** 3 good
**Contribution:** 3 good

**Summary:**

This paper studies bidding in repeated first-price actions with hints, which could be a point estimate or an interval. The authors first establish optimal regret bounds for both cases when the bidder's value is the same across auctions. The authors further show that when the private values could vary, the hints do not help much. The authors then examine the cases when the others' bids are sparse and demonstrate a better bound. The empirical results support and validate the theoretical findings.

**Questions:**

* It seems that for the sparsity assumption, even if it holds in practice now, it is vulnerable to future changes or adversarial attacks. In particular, the authors mention that "Part of the reason for the sparsity is due to the scenario where the maximum competing bid is the reserve price set by the seller." If the seller adopts the strategy to set randomized reserve, will the assumption break? Similarly, the competing bidders could adopt randomized bidding strategy as well.

**Limitations:**

The authors adequately addressed the limitations and potential negative societal impact is not applicable.

**Strengths And Weaknesses:**

Strength:
* The problem studied in this paper is timely, interesting and important.

* The paper is generally well-written, clear, and easy to follow.

* The results are novel and technically strong; and the empirical results nicely support and validate the effectiveness of the proposed methods.

Weakness

* The sparsity assumption requires more motivations and support from practice.

---

> ### Author Response · Authors · 2022-08-02
> **Response to reviewer Bere**
>
> We thank the reviewer for the thoughtful review and positive feedback. We use the sparsity assumption for two reasons: first, our minimax lower bounds in Section 3.4 give pessimistic results without any structural assumptions on the auction, while sparsity could alleviate this situation and lead to meaningful results; second, current empirical data mostly support the sparsity assumption. We do agree with the reviewer that bidding in first-price auctions should ultimately be modeled as a multi-agent game, instead of online learning for a single learner (bidder). However, from a technical perspective, this is a too complicated model and existing literature on first-price auctions mostly focused on a single bidder ([BGM+19, HZW20, HZF+20, BFG21]) to start with. From the practical perspective, many bidders have not yet well adapted their bidding strategy to the current environment, and using such an algorithm could exploit advantages in such an environment.

---

### Official Review · Reviewer_8AsX · 2022-07-12

**Rating:** 7
**Confidence:** 2
**Soundness:** 3 good
**Presentation:** 3 good
**Contribution:** 3 good

**Summary:**

In the past four years, an important development in the digital advertising industry is the shift from second-price auctions to first-price auctions. They consider a differentiated setup: assume that we have access to a hint that serves as a prediction of other bidders’ maximum bid, where the prediction is learned through some blackbox machine learning model. They consider two types of hints: one where a single point-prediction is available, and the other where a  hint interval (representing a type of confidence region into which others’ maximum bid falls) is available. They establish minimax optimal regret bounds for both cases and highlight the quantitatively different behavior between the two settings, and provide improved regret bounds when the others’ maximum bid exhibits the further structure of sparsity. They also complement the theoretical results with demonstrations using real bidding data.

**Questions:**

Line 259, what is CheW?

**Limitations:**

 I didn’t think it has negative societal impact.

**Strengths And Weaknesses:**

This paper studies first price auction with hint. Nowadays, first price auction obtains more research interest. The topic is important and the result is good. The writing is clear.

---

> ### Author Response · Authors · 2022-08-02
> **Response to reviewer 8AsX**
>
> We thank the reviewer for the nice summary and positive feedback. In Line 259, ChEW is the algorithm used in [HZF+20] which attains the $\widetilde{O}(\sqrt{T})$ regret without the hints; we will add appropriate pointers in the final version.

---

### Official Review · Reviewer_Bu8B · 2022-07-19

**Rating:** 7
**Confidence:** 4
**Soundness:** 4 excellent
**Presentation:** 3 good
**Contribution:** 3 good

**Summary:**

This paper studies the problem of online learning to bid in first-price auctions. In this problem, each round a bidder with some value for the item (either fixed over the duration of the bidding, or independently sampled from a known distribution) must submit a bid; simultaneously, a minimum-bid-to-win MBTW is generated (either adversarially or stochastically from the distribution). If the bidder bids above the MBTW, they win the item (and receive value - bid in utility); otherwise, they lose (and receive nothing).

This has been a popular question in recent years (largely motivated by the migration of several large advertising auctions from second-price to first-price); this work contributes to the literature by exploring what happens when the bidder is provided some side information regarding the true MBTW. This information takes the form of a “hint”; a value who is guaranteed to be somewhat close (on average) to the true MBTW. The authors differentiate between the cases where the learner receives just the hint each round (“single hint”), or the case where they receive the hint and how noisy it is (“hint interval”) each round. The ultimate regret bounds are in terms of the total error L of the hints (in a similar manner as data-dependent regret guarantees).

The authors show the following results:

- The authors characterize (up to logarithmic factors) the optimal regret guarantees for the single hint and hint interval setting, in the case where value is fixed. To do this, they present an adaptation of multiplicative weights that incorporates hint information. Interestingly, they show there exists a gap in regret guarantees between these two settings.
- They also characterize the optimal regret guarantees in the setting of varying private prices. They show in this case it is hard to take advantage of hint information to get asymptotically better regret guarantees.
- To address this, they investigate whether it is possible to get better regret guarantees if the MBTW is “sparse” and can take on at most K different values. Again they provide asymptotically tight regret guarantees in this setting.
- Finally, they perform empirical simulations of these learning algorithms on data taken from real auctions (and demonstrate they are effective at incorporating this side-information).


**Questions:**

In Algorithm 1, is it necessary to know all the accuracies sigma_t ahead of time? If not, how do you set the learning rate eta (it seem s like it depends on all sigma_t).

**Limitations:**

They have adequately addressed this.

**Strengths And Weaknesses:**

Understanding how to bid in first-price auctions is an immensely important problem in practice. Although this paper is not the first paper to study this, I think it makes some fairly interesting and significant contributions both theoretically (it is nice they have fairly tight characterizations of regret guarantees everywhere, and I think the existence of a gap between single hints and hint intervals is an interesting message) and experimentally (they present fairly simple / implementable methods for taking advantage of hint information, something which does exist in real life).

I am not super familiar with all the related literature, so it is hard for me to evaluate how novel these techniques are over e.g. the techniques in the related first-price bidding paper [HZF+ 20] or more general data-dependent learning papers like [WLA20]. My reading is that the proofs are technically involved but this paper doesn’t contribute any especially novel techniques (but I think this is OK in light of its other contributions).

Overall the paper was well-written and easy to read. I would recommend acceptance.

---

> ### Author Response · Authors · 2022-08-02
> **Response to reviewer Bu8B**
>
> We thank the reviewer for the detailed summary and positive feedback. We agree with the reviewer that our main contributions lie in the illustration of the separation between ``single hint" and ``hint intervals", as well as the new methodology to incorporate the hints in first-price auctions. The technical analyses of the algorithms and lower bounds are more or less standard.
>
> Algorithm 1: a slightly modified algorithm will have the same performance guarantee and require no knowledge of future $\sigma_t$. The idea is to change the current time-invariant learning rate $\eta$ into a decaying learning rate $\eta_t \propto (\sum_{s\le t} \sigma_s^{q/(q+1)})^{-1}$, and the analysis is similar. We will make this change in the final version of our manuscript.

---

### Author Response · Authors · 2022-08-02
**Response to all reviewers**

We are grateful to all reviewers for their careful reading and detailed reviews. In the rebuttal, we provide detailed point-to-point response to each and every comment of the reviewers, with a separate comment to each review.

---

### Meta-Review · Area_Chair_mbrZ · 2022-08-22

**Recommendation:** Accept
**Confidence:** Certain

**Metareview:**

The paper studies a regret minimization model of bidding in repeated first price auctions, when a noisy "hint" about the highest competing bid is available. The question is whether the availability of such a hint can significantly reduce the best achievable regret. The authors give almost matching lower and upper bounds on the regret in several cases (e.g., hint is provided as a point/interval estimate, or if the distribution of the highest competing bid has a small finite support). They also present experimental evaluation of the algorithms.

The reviewers agree that this is a practically relevant and theoretically interesting model, and the results are non-trivial and interesting. The proofs are technically involved, although they do not introduce particularly new techniques. While the main contribution of the paper is theoretical, the experimental results nicely complement the theory. Overall, this is a nice paper and can be accepted to NeurIPS.

**Award:**

No

---

### Decision · Program_Chairs · 2022-09-14

Accept